# Large-eddy simulation of an atmospheric bore and associated gravity wave effects on wind farm performance in the Southern Great Plains

Adam S. Wise[1], Robert S. Arthur[2], Aliza Abraham[3], Sonia Wharton[2], Raghavendra Krishnamurthy[4], Rob Newsom[4], Brian Hirth[5], John Schroeder[5], Patrick Moriarty[3], and Fotini K. Chow[1]

[1]Department of Civil and Environmental Engineering, University of California, Berkeley, California, USA
[2]Lawrence Livermore National Laboratory, Livermore, California, USA
[3]National Renewable Energy Laboratory, Golden, Colorado, USA
[4]Pacific Northwest National Laboratory, Richland, Washington, USA
[5]National Wind Institute, Texas Tech University, Lubbock, Texas, USA

**Correspondence:** Adam S. Wise (adamwise@berkeley.edu)

**Abstract.** Gravity waves are a common occurrence in the atmosphere, with a variety of generation mechanisms. Their impact on wind farms has only recently gained attention, with most studies focused on wind farm-induced gravity waves. In this study, the interaction between a wind farm and gravity waves generated by an atmospheric bore event is assessed using multi-scale large-eddy simulations. The atmospheric bore is created by a thunderstorm downdraft from a nocturnal mesoscale convective system (MCS). The associated gravity waves impact the wind resource and power production at a nearby wind farm during the American Wake Experiment (AWAKEN) in the U.S. Southern Great Plains. A two-domain nested setup ($\Delta x = 300$ m and 20 m) is used in the Weather Research and Forecasting (WRF) model, forced with data from the High-Resolution Rapid Refresh model, to capture both the formation of the bore and its interaction with individual wind turbines. The MCS is resolved on the large outer domain, where the structure of the bore and the associated gravity waves are found to be especially sensitive to parameterized microphysics processes. On the finer inner domain, gravity wave interactions with individual wind turbines are resolved; wake dynamics are captured using a generalized actuator disk parameterization in WRF. The gravity waves are found to have a strong effect on the atmosphere above the wind farm; however, the effect of the waves is more nuanced closer to the surface where there is additional turbulence, both ambient and wake-generated. Notably, the gravity waves modulate the mesoscale environment by weakening and dissipating the pre-existing low-level jet, which reduces hub-height wind speed and hence the simulated power output, which is confirmed by the observed supervisory control and data acquisition (SCADA) power data. Additionally, the gravity waves induce local wind direction variations correlated with fluctuations in pressure, which lead to fluctuations in the simulated power output as various turbines within the farm are subjected to waking from nearby turbines.

# 1 Introduction

The United States (U.S.) Southern Great Plains, and more specifically, Oklahoma, is a region with significant wind turbine and wind farm development. Oklahoma currently has nearly 12 GW of installed wind capacity and the state has recently generated over 44% of its electricity demand from wind power, enough to power 3.3 million homes (U.S. Energy Information Administration, 2023). The region has abundant wind energy resources and is an area that will see continued wind farm development; it is critical to study atmospheric-wind farm interactions in this region further. In the Southern Great Plains, there are a number of relevant atmospheric science phenomena for which the effect on wind turbines and wind farms is not well understood. One such phenomenon, which is of particular interest to the wind energy community as part of the ongoing American Wake Experiment (AWAKEN) (Moriarty et al., 2020; Debnath et al., 2022; Moriarty et al., 2024), is intermittent or wavy turbulence.

In general, wavy turbulence in the atmosphere can either be shear-driven or buoyancy-driven. For shear-driven waves, competing shear and buoyancy driven effects result in an instability (Stull, 1988) thus inducing waves. Shear-driven waves commonly occur beneath low-level jets (LLJs), which exhibit a large amount of shear, and are dependent on local atmospheric conditions (Newsom and Banta, 2003). The occurrence of shear-driven waves in the U.S. Great Plains can depend on local topography, such as shallow river valleys, or land-use variability that affects wind shear profiles via changes in surface roughness. Buoyancy-driven waves, also known as gravity waves, generally occur due to perturbations in a stably stratified atmosphere (Stull, 1988). Gravity wave generation mechanisms include frontal systems, thunderstorms, and mountains (Rottman and Simpson, 1989; Ralph et al., 1999; Stull, 1988; Geerts et al., 2017), or even the wind farms themselves (Allaerts and Meyers, 2018; Lanzilao and Meyers, 2022; Stipa et al., 2024). Waves generated by mountains and wind farms are typically standing waves, while waves generated by frontal systems and thunderstorms propagate with a wave speed that depends on atmospheric conditions.

In this study, we focus on gravity waves associated with atmospheric bore events, which are commonly observed in the U.S. Great Plains. Bores are often generated by a nocturnal mesoscale convective systems (MCS) (Haghi et al., 2019; Feng et al., 2019). In this case, a cold air pool created by the thunderstorm downdraft spreads radially as a density current resulting in a bore (Rottman and Simpson, 1989; Toms et al., 2017; Johnson et al., 2018; Haghi et al., 2019; Haghi and Durran, 2021) (although, note that density currents can also be generated by distant cold fronts, atmospheric mesoscale disturbances, or rapid surface cooling (Simpson, 1997; Sun et al., 2002; Lundquist, 2003)). Bores result in a temporary increase in the depth of the stable boundary layer (SBL), along with wavy oscillations and a shift in wind speed and wind direction associated with the propagation of the bore (Knupp, 2006).

Bore events were a major focus of the Plains Elevated Convection at Night (PECAN) Field Campaign, which took place during June and July 2015 (Geerts et al., 2017). PECAN set out to study nocturnal deep convection using a number of instruments, including mobile Doppler lidars, radars, atmospheric emitted radiance interferometers, radiosondes, and aircraft observations. Six intensive observational periods (IOPs) and two unofficial field operations were dedicated to bores, although bores were also

observed during IOPs focused on other processes (Geerts et al., 2017; Weckwerth et al., 2019; Weckwerth and Romatschke, 2019).

In addition to the data gathered to characterize bores, forecasting and modeling of bores was also a critical component of the field campaign. Johnson et al. (2018) and Johnson and Wang (2019) conducted a number of Weather Research and Forecasting (WRF) model evaluations for predicting bores during PECAN. In their studies, they initialized simulations using the gridpoint statistical interpolation-based ensemble Kalman filter technique, where both in situ and convective-scale radar data are assimilated (Johnson and Wang, 2017; Johnson et al., 2017). They found that bore structure is especially sensitive to microphysics parameterizations, as microphysical low-level cooling within convection determines the strength of the cold pool. Other studies focused on deep convection have also found large modeling sensitivities to microphysics schemes (Pandey et al., 2023; Han et al., 2019). Johnson and Wang (2019) assessed the sensitivity of model predictions to the horizontal and vertical grid spacing, as well as the turbulence closure. They found that 250 m grid spacing offered improved results compared to 1-km grid spacing, with further improvement when using large-eddy simulation (LES). Additionally, enhanced low level vertical resolution improved simulation results. For additional discussion on nocturnal convection initiation and the forecastability of MCSs and bores, in general, see Weckwerth and Romatschke (2019) and Weckwerth et al. (2019).

In the present work, LES is used to examine how an atmospheric bore and its associated gravity waves affect wind farm performance. LES explicitly solves for the most energetic turbulent eddies in the atmospheric boundary layer (ABL) while parameterizing the effects of the smaller turbulent length scales on the resolved-scale flow. LES is especially well suited to capture transient and dynamic turbulent flow structures, such as bores, which are important features of the ABL that interact with wind farms. We use a nested multi-scale framework in the WRF model v4.4 (Skamarock et al., 2021), which allows us to explicitly resolve the convective-scales that generate the bore. The impact of the bore on the wind farm is simultaneously parameterized using a generalized actuator disk (GAD) to resolve the finer-scale turbulence associated with wind turbine wakes (Mirocha et al., 2014; Aitken et al., 2014). The WRF-LES framework along with the GAD parameterization is hereinafter denoted as WRF-LES-GAD to describe the model in its entirety.

The goals of this study are to: (1) design a multi-scale framework to analyze bore-wind farm interactions, (2) analyze the sensitivity of the bore structure to modeling parameters, and (3) analyze how an intermittent turbulence event such as from a bore can modulate wind farm power production. To accomplish these goals, we use WRF-LES-GAD to model the King Plains wind farm in Oklahoma, which is a major focus of the AWAKEN field campaign (described in more detail in Sect. 2). The WRF-LES-GAD modeling setup is described in detail in Sects. 3 and 4, focusing on the microphysics schemes and LES turbulence closures which are known to affect model performance. Observations of the atmospheric bore and its characteristics at various times are compared with modeling results in Sect. 5. Lastly, the effect of the gravity waves on the simulated wind farm's power production is also quantified in Sect. 5 by analyzing time periods before, during, and after the gravity waves pass the Kings Plains wind farm.

## 2 Case study

### 2.1 Overview of the AWAKEN field campaign

The American WAKE experimeNt (AWAKEN) field campaign began in September 2022 with a scheduled end date of September 2025. AWAKEN is centered around five wind farms in northern Oklahoma near the town of Enid. While the AWAKEN domain covers a number of wind farms, the bulk of the instrumentation is located in the eastern half of the King Plains wind farm, which includes 50 wind turbines (see Fig. 1). The aim of the field campaign is to better understand wind farm-atmosphere interaction, and as part of AWAKEN, there are seven testable hypotheses (Moriarty et al., 2024). The focus of this study is on the hypothesis that intermittent turbulent bursting events related to Kelvin-Helmholtz instability, gravity waves, and bores lead to fluctuations in wind farm power production and structural loading of wind turbines. Specifically, this study is concerned with the effect of bores and associated gravity waves on wind farm power production.

The dominant wind direction in the region is southerly (Krishnamurthy et al., 2021; Debnath et al., 2023), which determined the design of a north-south instrument transect through the King Plains wind farm. In this study, we focus on data collected with remote sensing Doppler lidars at the A1 location (see Fig. 4). A scanning Doppler lidar (Halo Streamline XR+) deployed by the Atmospheric Radiation Measurement (ARM) (Newsom and Krishnamurthy, 2022) ran composite scans for 20 minutes of six-beam profiling (Sathe et al., 2015) and 10 minutes of vertical stares. The scanning Doppler lidar has a range gate of 30 m and measured from 90 m to the top of the ABL with a temporal resolution of approximately 6 seconds per wind profile. The six-beam profiles are used for comparison in this study, with a least squares fit to the radial velocity measurement (Krishnamurthy et al., 2024). At Site A1, an additional short-range vertical profiling Doppler lidar was deployed by Lawrence Livermore National Laboratory (LLNL) to focus on the wind turbine rotor layer (40 m to 240 m). This higher-resolution lidar at A1 was a Windcube v2, which has a scan frequency of approximately 4 seconds.

This study also includes measurements generated by two X-Band radars from Texas Tech University (TTU) (Hirth et al., 2017; Debnath et al., 2022; Hirth et al., 2024). These radars provide wind measurements over a large portion of the study region (see Fig. 1), performing 145 degree horizontal scans over a vertical range of 2 degrees, resulting in observations over a three-dimensional volume roughly every two minutes. Dual-Doppler reconstruction is used to obtain the horizontal velocity vector field from the radial wind speeds recorded by the radars. In their near range, the radars provide observations with a grid spacing as fine as 25 m. However, the eastern half of the King Plains wind farm, which is the focus of this study, is on the edge of the maximum radar range, where the radar measurements are interpolated to a grid spacing of 50 m. Additionally, it is worth noting that there are areas without returns within the rotor swept area due to the lower elevation of the eastern half of the King Plains wind farm as the radar beam is blocked by terrain variations. There is also the impact of the Earth's curvature where at ranges of 20, 25, and 30 km, curvature accounts for shifts in vertical height of 49, 70, and 96 m, respectively.

There are surface-based meteorological (met) stations scattered throughout the AWAKEN domain, which are used to characterize atmospheric surface layer stability. At Site A1, an eddy correlation flux measurement system (ECOR) was deployed from April until September 2024 (Cook, 2018). The ECOR system includes a sonic anemometer mounted 3 m above ground

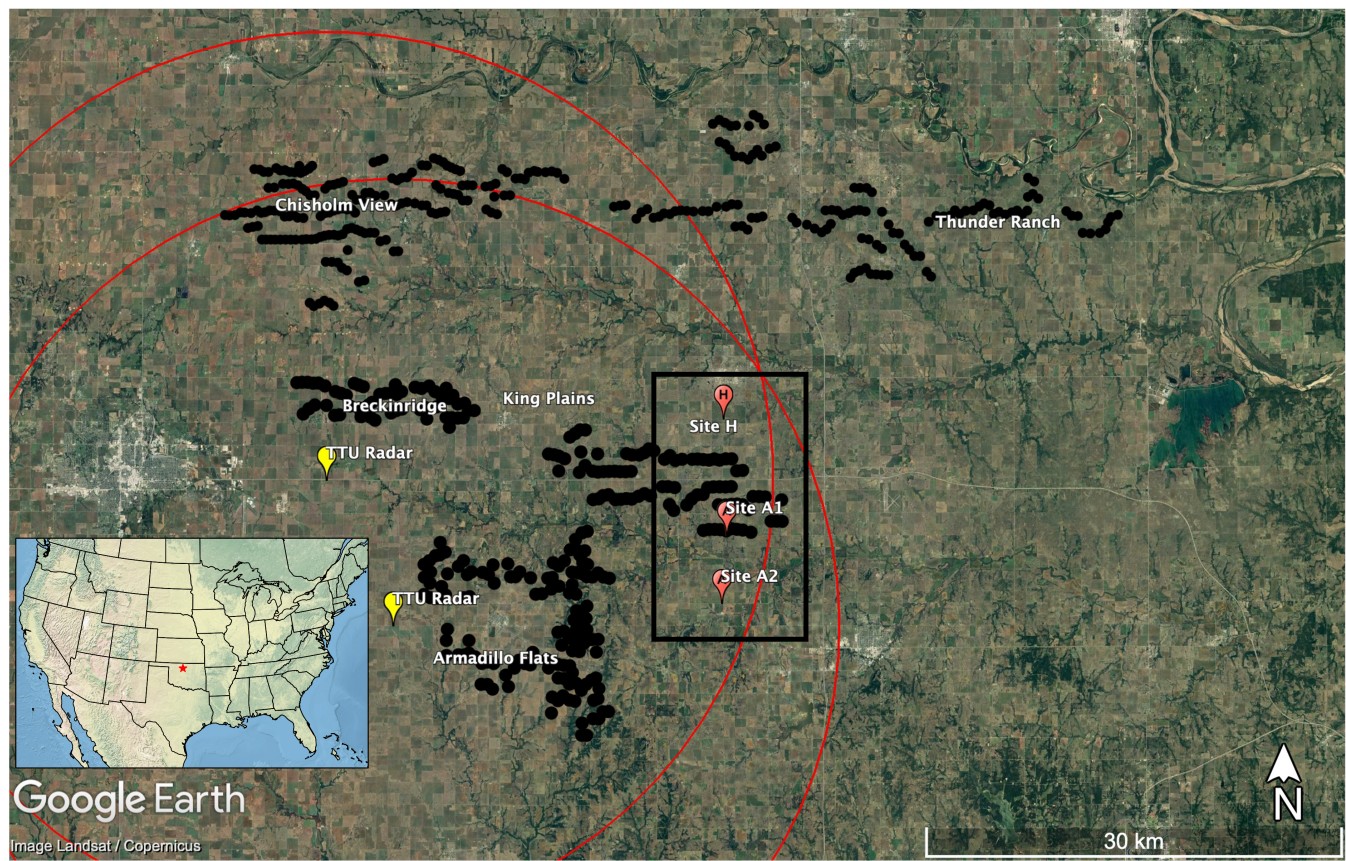

**Figure 1.** AWAKEN region with instrumented sites relevant to this site and black dots representing individual turbines. The red star on the inset map of the continental U.S. is the location of the AWAKEN region within the state of Oklahoma. The X-band radars' ranges are highlighted with red circles. The area representing WRF domain d02 is outlined in black and covers the eastern half of the King Plains wind farm (see Fig. 4). © Google Earth.

level (a.g.l.), measuring three-dimensional wind at 10 Hz. The data are post-processed in near-real-time into 30 minute fluxes. Fast-response wind and temperature measurements from the sonic anemometer are used to derive the Obukhov length.

Lastly, high-frequency power output data at 1 Hz from the supervisory control and data acquisition (SCADA) system are also available for the King Plains wind farm. Simulated and observed power are qualitatively compared in Sect. 5.4 to demonstrate the dynamic effects of the bore and associated gravity waves on power output. The observed power data are normalized at the request of the wind farm operator.

## 2.2 Case study description

The phenomenon of interest in this work is an atmospheric bore with associated gravity waves. The historical weather radar imagery indicates that the weather consisted of localized precipitation on 06 June 2023. Figure 2 shows the weather radar

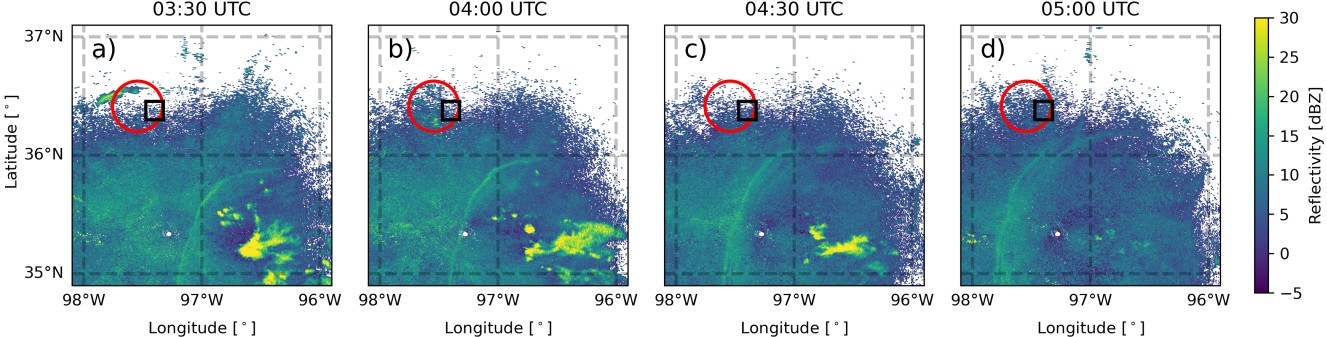

**Figure 2.** NEXRAD WSR-88D radar reflectivity over central Oklahoma at (a) 03:30, (b) 04:00, (c) 04:40, and (d) 05:00 UTC on 06 June 2023. The radar returns are from the Oklahoma City Site (KTLX). The area representing WRF domain d02 is outlined in black (see Fig 4) and the approximate return from the TTU X-Band radars are outlined in red. An animation of the NEXRAD data from 03:30 - 06:00 UTC (22:30 - 01:00 local time) is included in the supplementary material (see Video 1, in the Video Supplement).

reflectivity from the NEXRAD (NEXt generation of RADar) Weather Surveillance Radar 88D (WSR-88D) system at the Oklahoma City radar site (KTLX) operated by the National Weather Service. The radar images in Fig. 2 are from 03:30 to 05:00 UTC (22:30 to 00:00 local time) in 30 minute intervals. However, the radar returns have a roughly 4 minute temporal resolution and animation of the event is available in the Video Supplement. Localized precipitation caused by a cluster of
thunderstorms is evident as increased reflectivity in Fig. 2. These storms form an organized nocturnal mesoscale convective system (MCS). At 03:30 UTC, the cold pool outflow from the MCS is evident as a fine line to the north and west of the MCS center. Similar outflow boundaries are commonly observed along with convective systems (Markowski and Richardson, 2010; Houze Jr., 2004). Over time, the outflow boundary propagates radially away from the MCS, approaching the AWAKEN region to the northwest.

The cold pool outflow visible as a fine line in Fig. 2 and as similarly seen for another MCS by Tomaszewski and Lundquist (2021). The fine line in this study represents the front of the atmospheric bore that ultimately reaches the King Plains wind farm. The TTU X-Band radars measure the wind speed over a region encompassing the AWAKEN site, with Fig. 3 showing wind speeds at various heights around 06:30 UTC. Starting at approximately 06:00 UTC on 06 June 2023, oscillations in the wind speed measurement were observed by the X-band radars (see Video 2 in the Video Supplement). The oscillations
passed through the entirety of the King Plains wind farm at approximately 06:30 UTC. The plan view shown in Fig. 3 has been post-processed to terrain-following heights of 95 m (which is close to the 88.5 m turbine hub-height), 145 m and 270 m a.g.l.

    The radar observes bands of faster and slower wind speeds indicative of wavy turbulence (Fig. 3). In Fig. 3(a), at 95 m a.g.l. (near hub-height) the wave pattern is not clear, likely because the waves interact with the wind farm and ambient turbulence closer to the surface. However, at 270 m a.g.l. (the highest altitude measured by the radars), the oscillations, seen as alternating
color bands stretching left-right across the image, are much more distinct (Fig. 3(c)). While the gravity waves become weaker

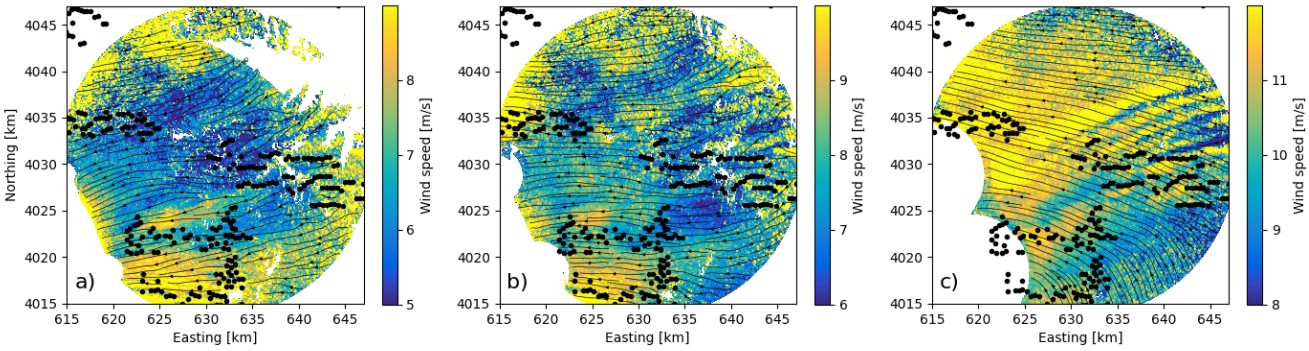

**Figure 3.** Plan view of Dual-Doppler wind speed magnitude and streamlines from the TTU X-Band radars at 06:27:57 UTC on 06 June 2023 at heights of (a) 95 m, (b) 145 m, and (c) 270 m a.g.l. The time-stamp corresponds to the start of the scan routine completed by the radars which take approximately two minutes for a full return. An animation of the TTU X-Band radar data from 05:30-07:30 UTC is included in the supplementary material (see Video 2, in the Video Supplement).

closer to the surface, they are faintly visible in Fig. 3(b), which represents a height close to the top of the wind turbine rotor layer.

Prior to the bore passage, the ambient atmosphere in the study region can be characterized as stable, as is typical for nighttime conditions in Oklahoma. 06:30 UTC corresponds to 01:30am central daylight time (CDT) with sunset occurring nearly five hours prior at 08:43pm CDT. The Obukhov length from the surface met station at A1 is 13.8, -27.9, -23.4 m at 06:00, 06:30, 07:00 UTC, respectively, indicating very stable conditions prior to the bore followed by unstable conditions. The passage of the bore perturbs the stable boundary layer, inducing gravity waves that are eventually seen in the measured wind speed measurements in Fig 3. While the thermodynamic profiling instrumentation from AWAKEN was not online during this specific event, modeling results shown later in this study (see Fig. 6) suggest that the vertical structure of the atmosphere is stably stratified prior to the bore with additional discussion on how the bore affects the thermal structure in Sect. 5.3.

As denoted by the streamlines in Fig. 3, the ambient hub-height wind direction is easterly, a non-ideal wind direction for the King Plains wind farm as the layout was designed for the more dominant southerly wind direction. Interestingly, the gravity waves propagate from the south, which has a strong influence on the hub-height wind direction as discussed in later in this study. Additionally, the hub-height wind speeds are relatively low, on the order of 4-6 m s$^{-1}$, such that the power output of the farm is highly susceptible to wind speed fluctuations in this range (in region 2 of the power curve or in the operational power-maximizing control mode of the wind turbines).

## 3  Methods

### 3.1  WRF-LES-GAD

The present work uses the large-eddy simulation capability of the WRF model, version 4.4, with modifications including the generalized actuator disk (GAD) (Mirocha et al., 2014) with a turbine yawing capability (Arthur et al., 2020); a stochastic inflow perturbation method (the cell perturbation method; CPM) (Muñoz-Esparza et al., 2014, 2015); and the implementation of a dynamic turbulence closure (Chow et al., 2005; Kirkil et al., 2012). The GAD requires specifications for the turbine's airfoil lift and drag coefficients. However, the required lift and drag parameters for the 2.8-MW General Electric turbines installed at King Plains are not publicly available. We therefore use the open source 2.8-MW turbine developed by the National Renewable Energy Laboratory (NREL) as a suitable representation in the model, with details available in Quon et al. (2024). The turbine has a hub-height of 88.5 m and rotor diameter of 126 m. Minor differences between the NREL turbine and the actual turbines at King Plains are not expected to be critical to the conclusions of this study.

### 3.2  Microphysics parameterizations

In modeling studies involving deep convection, simulation results show sensitivity to the microphysics parameterization (Pandey et al., 2023; Han et al., 2019). Microphysics schemes represent cloud and precipitation processes, describing the formation and growth of water particles (hydrometeors) for clouds, which are especially relevant to the MCS and bore formation in this study. The representation of microphysics in atmospheric models that are cloud-resolving is a major source of uncertainty and an active research area (Morrison et al., 2020; Tatsuya Seiki and Satoh, 2022). The microphysics schemes explored in this study in order of increasing sophistication are the Thompson (Thompson et al., 2008), WRF double-moment 6-class (WDM6) (Lim and Hong, 2010), and Morrison (Morrison et al., 2009) schemes, which are widely used by both operational models and in research. Microphysics schemes largely fall into two categories: single-moment schemes that predict the mixing ratios of hydrometeors and double-moment schemes (WDM6 and Morrison) that predict both the mixing ratio and the number concentration of hydrometeors. The schemes used in this study vary in what prognostic variables or species of particles they predict, which are detailed in Table 1. Note that some double-moment schemes may only be double-moment for a limited number of particle species, as is the case for the Thompson model, which is why the Thompson model is denoted as 1.5-moment in Table 1. More detail on WRF microphysics schemes can be found in Skamarock et al. (2021).

### 3.3  Turbulence modeling

Extensive literature in LES turbulence closure modeling has shown that dynamic turbulence closures are able to perform better than standard eddy-viscosity approaches in SBL conditions (Zhou and Chow, 2011, 2014; Wise et al., 2024). The standard approach to turbulence modeling relies on an implicit filter related to the grid resolution to parameterize the effect of the subgrid scale (SGS) motions on the resolved flow. In this study, we use an explicit filtering approach to separate large-scale

**Table 1.** Description of various microphysics schemes used in WRF-LES-GAD.

| Parameterization (Year published) | Sophistication | mp_physics | Variables |
|---|---|---|---|
| Thompson (2009) | 1.5-moment | 8 | $Qc, Qr, Qi, Qs, Qg, Ni, Nr$ |
| WDM6 (2009) | Double-moment | 16 | $Qv, Qc, Qr, Qi, Qs, Qg, Nn, NcNr$ |
| Morrison (2010) | Double-moment | 10 | $Qc, Qr, Qi, Qs, Qg, Nr, Ni, Ns, Ng$ |

Mixing ratio of water vapor ($Qv$); mixing ratio and number concentration of cloud water ($Qc, Nc$), rain ($Qr, Nr$), cloud ice ($Qi, Ni$), snow ($Qs, Ns$), and graupel ($Qg, Ng$); and number concentration of cloud condensation nuclei ($Nn$).

from subfilter scale motions which provides a framework for using the dynamic procedure of Germano et al. (1991) to solve for coefficients of interest.

When an explicit filter is used, the presence of the numerical grid divides the subfilter scale (SFS) motions into resolved and unresolved portions. The unresolved SFS motions are the commonly referred to SGS motions. However, the effect of resolvable subfilter scale (RSFS) motions can be reconstructed using a scale-similarity approach (see Chow (2004) for derivations). The RFSF motions are neglected in standard closure schemes (i.e., those that do not use an explicit filter). In this study, we make use of the explicit filtering and reconstruction approach of the Dynamic Reconstruction Model (DRM) turbulence closure (Chow et al., 2005). The performance of the DRM closure with respect to ambient turbulence for this specific case study is compared with other closures in Appendix A.

## 4 Model configuration and description

The two-domain nested setup for WRF-LES-GAD is shown in Fig. 4 with details in Table 2. The setup is unique compared to other multi-scale WRF setups in that it uses only two domains, which makes it very computationally efficient. Mesoscale forcing is provided by the High-Resolution Rapid Refresh (HRRR) model v4 (Dowell et al., 2022) as the lateral boundary conditions (updated hourly) for domain d01 following the procedure of Blaylock et al. (2017). The HRRRv4 model was chosen because its horizontal grid spacing is 3 km, which is much finer than the grid resolution used by other regional or global models. In comparison, the Global Forecast System (GFS) (National Centers for Environmental Prediction, National Weather Service, NOAA, U.S. Department of Commerce, 2015) and the European Centre for Medium-Range Weather Forecasts Reanalysis v5 (ERA5) (Hersbach et al., 2020) have grid spacings of 0.25 degrees, which corresponds to roughly 27-28 km. The HRRRv4 3 km grid spacing is already a very fine-scale mesoscale simulation, and using a relatively large parent grid ratio of 10 from HRRRv4 to domain d01 is reasonable to intentionally skip across the convective gray zone where turbulence is only partially resolved (Wyngaard, 2004; Chow et al., 2019; Haupt et al., 2019; Muñoz-Esparza et al., 2017).

The HRRRv4 model was also chosen because of its high temporal update frequency and use of data assimilation (DA). The temporal update is hourly, which is important for modeling dynamic events. ERA5 has an hourly temporal update; the GFS has a temporal update of three hourly, with forecasts initialized every six hours. In HRRRv4, Next Generation weather radar (NEXRAD) data is assimilated every 15 minutes during the first hour of simulation, which is referred to as the pre-forecast

hour (Dowell et al., 2022). Preliminary simulations were also conducted using GFS and ERA5, which resulted in nocturnal MCSs but in the incorrect location (not shown). In this study, we use the HRRRv4 analysis product which includes assimilated NEXRAD data. Simulations showed that the DA in the HRRRv4 model constrains the initiation of the MCS to the correct location, which is critical to the present study.

Domain d01 is centered over the approximate midpoint between the AWAKEN region and the location where the MCS initiates. The 300 m horizontal grid spacing was chosen as a balance between computational cost and model resolution over a large geographic area. Note that preliminary simulations at 1 km grid spacing did not provide realistic bore structure (not shown). Additional preliminary simulations at 200 m grid spacing provided similar results as those with the 300 m grid spacing (not shown), which is consistent with the findings of Johnson and Wang (2019) who found little benefit to increasing the grid resolution beyond 250 m for their bore simulations. As previously mentioned, modeling studies that resolve deep convection show large sensitivities to cloud and precipitation processes. Therefore, three separate simulations for domain d01 are run with the microphysics schemes specified in Table 1 with results compared in Sect. 5.1.

Domain d02 has 20 m horizontal grid spacing, which is necessary to resolve the effects of individual wind turbines using the generalized actuator disk parameterization. The 20 m horizontal grid spacing is fine enough to include 6 grid points across each generalized actuator disk in the horizontal (and 16 in the vertical with the grid spacing described in Table 2). The GAD parameterization is typically used at 10 m spacing (Mirocha et al., 2014; Arthur et al., 2020; Wise et al., 2022), but given the large domain size required here, preliminary simulations showed that 20 m horizontal grid spacing provides reasonable results. When using 20 m grid spacing, and prior to the turbulent mixing in the far-wake region, the near-wake structure importantly still retains the characteristic bimodal distribution of the velocity deficit (also described as a double-Gaussian velocity deficit in Keane et al. (2016) and Schreiber et al. (2020)). The bimodal distribution is due to blade geometry and aerodynamics, as well as nacelle effects as seen in experiments, observations, and modeling (Wang et al., 2017; Vermeer et al., 2003; Carbajo Fuertes et al., 2018).

A parent grid ratio of 15 from d01 to d02 bridges the MCS-resolving domain (domain d01) to the turbine wake-resolving domain (domain d02). Intermediate nests with grid spacing between 300 and 20 m were explored but found to lessen the agreement with observations (not shown) as in Mazzaro et al. (2017). Because of the large parent grid ratio between domains d01 and d02, the development of small-scale turbulent structures on d02 is accelerated using the cell perturbation method (CPM), a stochastic inflow perturbation method (Muñoz-Esparza et al., 2014, 2015). The CPM works by applying small temperature perturbations at the domain boundaries, fostering the development of a wider range of turbulent scales with negligible computational cost. CPM has been used successfully in a large number of WRF-LES studies (Arthur et al., 2020; Connolly et al., 2021; Wise et al., 2022; Sanchez Gomez et al., 2022, 2023).

At 20 m grid spacing, much of the ambient turbulence is resolved on d02; however, SBL conditions tend to reduce the turbulent length scale. Previous work has shown that dynamic turbulence closures can resolve more ambient turbulence compared to standard closures at coarser grid resolutions, especially in strongly stable conditions (Zhou and Chow, 2011, 2014; Wise et al., 2024). Therefore, the DRM closure is especially well-suited for this case study and simulation setup. The sensitivity of resolved turbulence characteristics to the turbulence closure is explored in Appendix A.

**Table 2.** Parameters used for the nested multi-scale WRF-LES-GAD setup. For the vertical resolution, $\Delta z_{min}$ is for the first grid point above the surface and is approximate due to the nature of the terrain-following coordinate system in WRF.

| Domain | $\Delta x$ [m] | Grid ratio | ~$\Delta z_{min}$ [m] | $N_x \times N_y$ | $\Delta t$ [s] | turb. closure |
|--------|------|------------|------------|------------------|-------|---------------|
| Forcing | 3000 | - | 15 | $1800 \times 1060$ | 20 | MYNN-EDMF |
| d01 | 300 | 10 | 8 | $1200 \times 1200$ | 1.5 | TKE-1.5 |
| d02 | 20 | 15 | 8 | $661 \times 871$ | 0.1 | DRM |

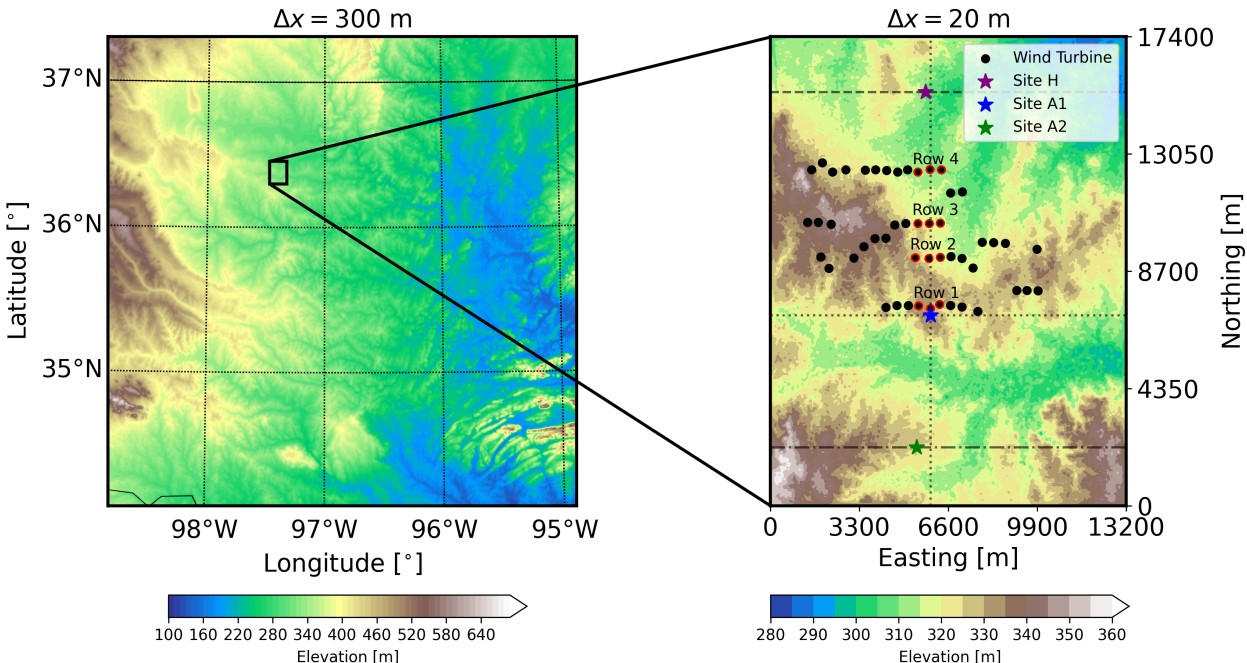

**Figure 4.** Topography of domains (a) d01 and (b) d02 used in the multi-scale simulation. Dimensions of each domain and other configuration information are included in Table 2. Note that the two plots have different ranges for their colorbars due to the maximum and minimum elevations within each domain.

Because of the computational expense of these simulations, the start time of domains d01 and d02 is staggered. Domain d01 begins on 06 June 2023 at 03:00 UTC, which is just over 3 hours prior to the gravity wave reaching the measurements at the AWAKEN site. A 3-hour lead time is chosen following the favorable results from the forecast start-time sensitivity conducted by Johnson and Wang (2017). Domain d02 begins at 05:00 UTC and is run until 07:30 UTC with the first hour (05:00-06:00 UTC) considered to be spin up, and therefore not used in the analysis. With both domains running concurrently, 30 minutes of simulation time takes approximately 24 hours of wall-clock time on 800 cores.

WRF-LES-GAD is run using a third-order Runge-Kutta time advancement scheme, with fifth-order horizontal and third-order vertical advection schemes. The physical parameterizations are as follows: the Noah land surface model (Chen and Dudhia, 2001), the Rapid Radiative Transfer Model for longwave radiation (Mlawer et al., 1997), and the Dudhia shortwave radiation model (Dudhia, 1989). No cumulus parameterization option is used as convection is explicitly resolved using LES.

For the turbulence closure, on domain d01, we use the standard turbulent kinetic energy 1.5-order (TKE-1.5) LES closure (Deardorff, 1980). On domain d02, where ambient turbulence is resolved and the wind turbines are parameterized, we use the DRM LES closure (with comparisons using other closures shown in Appendix A). All domains use the Eta similarity surface layer scheme (Janjić, 1994), which uses similarity theory (Monin and Obukhov, 1954) to determine the relevant surface fluxes. For topography, high-resolution terrain data (1-arc-second, approximately 30 m) are used from the Shuttle Radar Topography Mission (Farr et al., 2007). The landuse data similarly uses 30 m spacing and is a National Land Cover Database product converted into USGS categories (as in Chen et al. (2024)). The domain extends 20 km above the ground; at the upper-boundary condition, diffusive damping is applied with a coefficient of 0.01 following Johnson et al. (2018).

## 5   Results and discussion

### 5.1   Bore structure sensitivity to microphysics parameterizations

The case study examined in the present work is an atmospheric bore event with associated gravity waves that were observed to interact with the King Plains wind farm on 06 June 2023. The development of the bore is highly sensitive to the selected microphysics parameterization. Figure 5 shows the vertical velocity at 1 km a.g.l. and the potential temperature at 200 m a.g.l. for 04:00 UTC, which is an hour after the simulation/forecast start time. All three microphysics parameterizations resolve the deep convection and look qualitatively similar. Subtle differences in the strength of the cold pool, however, ultimately affect the development of the bore as it propagates roughly 100 km to the AWAKEN region.

The cold pool generated by the MCS is governed by latent cooling from the downdraft generated due to precipitation. The precipitation partially evaporates which produces the colder air downdraft, which forms the cold pool (Markowski and Richardson, 2010; Muller and Abramian, 2023). The cold pool then spreads horizontally as a density current when it reaches the surface. Planar-averaged vertical profiles of potential temperature and various hydrometeor species within the MCS are shown in Fig. 6. The subdomain that encompasses the MCS is shown in Fig. 5. The thermal structure of the MCS is similar for the three different microphysics parameterizations; however, results with the WDM6 microphysics are colder from the surface up to 3 km a.g.l. The stronger cold pool for the WDM6 microphysics scheme compared to the other schemes is because below the cloud base (approximately 2 km a.g.l., see Fig. 6(b)), the rain water mixing ratio decreases with decreasing altitude. The decrease in rain water mixing ratio indicates low-level rain evaporation as discussed by Johnson et al. (2018), which provides latent cooling. Both the Thompson and Morrison microphysics schemes have relatively constant rain water mixing ratios below the cloud base. Another contributing factor to the differences in the thermal structure of the MCS is the larger snow water mixing ratios in the upper atmosphere above the cloud base, which, in-turn, reduce the graupel and rain water mixing ratios. These trends are similar to those found by Johnson et al. (2018) in their microphysics sensitivity study.

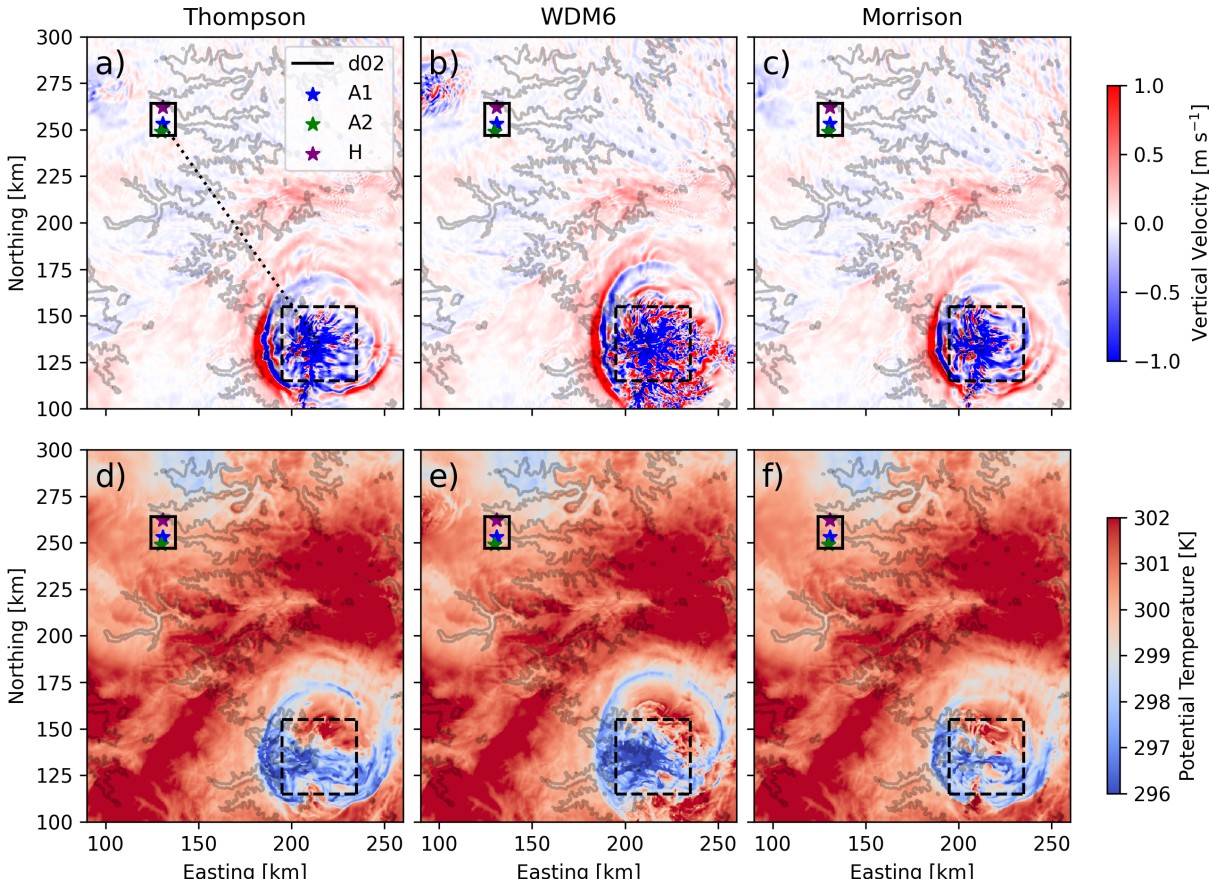

**Figure 5.** Instantaneous plan slice of (a-c) vertical velocity at 1 km a.g.l. and (d-f) potential temperature at 200 m a.g.l. on domain d01 for various microphysics schemes. Line contours represent 100 m changes in elevation. Note that the domain has been cropped for visualization purposes. A subdomain is denoted as the dashed box, which is used for analysis in Fig. 6. The dotted line in (a) is a transect used for the visualization in Fig. 9.

In contrast with Johnson et al. (2018), only the WDM6 microphysics scheme provides realistic bore structure by the time it reaches the AWAKEN region. Time-height contours in Fig. 7 show vertical profiles of wind speed and vertical velocity for the three different microphysics parameterizations compared with the lidars at A1. Note that the observations in Figs. 7(a) and 7(b) include both the Halo XR+ lidar, which measures from 100 m a.g.l. to the top of the boundary layer but with an alternate scan schedule every 20 minutes, as well as the Windcube v2, which focuses on lower altitudes measuring from 40 to 240 m a.g.l. While there are faint wavy structures in the simulation results using Thompson or Morrison, only WDM6 provides gravity waves that resemble observations and is therefore selected for further analysis. With WDM6, the period of the initial wave is ∼5 minutes in the model compared to ∼4.5 minutes for the observations as defined by the time between the first two wind speed peaks. There is a 16 minute delay in the timing of the bore for the results with WDM6 compared to observations,

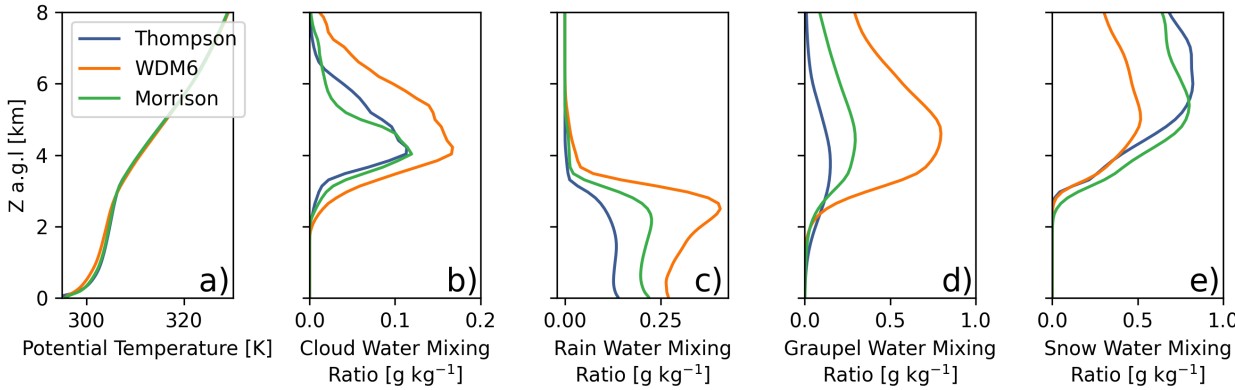

**Figure 6.** Vertical profiles averaged in the subdomain from Fig. 5 for (a) potential temperature, (b) cloud water mixing ratio, (c) rain water mixing ratio, (d) graupel water mixing ratio, and (e) snow water mixing ratio for the various microphysics schemes.

likely due to small differences in the bore speed with the gravity waves first appearing at 06:09 UTC in observations compared to 06:25 UTC in the model. Considering that the bore propagates over 100 km from the MCS to the AWAKEN region, for example, a difference in the bore speed of 1 m s$^{-1}$ can result in a timing difference of nearly 20 minutes. Additionally, the bore propagation speed is not necessarily constant with time as it can be affected by the ambient environment (Johnson and Wang, 2017; Haghi et al., 2019).

The mesoscale environment is very similar regardless of microphysics scheme. Figure 7 shows that a low-level jet (LLJ) exists prior to the bore passage. The maximum wind speed in the LLJ is predicted accurately by the model at 270 m a.g.l. as was observed; however, there is a positive bias in the wind speeds predicted by the model. These differences can be quantified by time-averaging the flow prior to the gravity waves (05:30-05:50 for the observations and 05:46-06:06 for the model accounting for the time delay). Specifically, for results using WDM6, the model predicts a LLJ maximum wind speed of 10.2 m s$^{-1}$ whereas the observed maximum wind speed by the Halo XR+ lidar is 9.3 m s$^{-1}$. During the same time period, the model predicted hub-height wind speed is 5.5 m s$^{-1}$ whereas the observed hub-height wind speed by the Windcube v2 lidar is 4.7 m s$^{-1}$ These errors in the wind speed are likely because of differences in the HRRRv4 forcing used as the lateral boundary conditions. Importantly, the wind direction at hub height is well-predicted by the model at 54.5 degrees whereas the observed wind direction is 51.4 degrees. Notably, the LLJ is weakened after the bore passage, which is discussed in more detail later.

The vertical velocity and wind speed signals in Fig. 7 are highly correlated during the gravity wave passage. During an updraft or downdraft of the wave, there is a corresponding convergence or divergence in the horizontal velocity. In Figs. 7(e) and 7(f), there are two strong peaks followed by residual wavy structures with less intensity. In the observations, there are three strong peaks which are followed by a 10 minute period of missing observations due to an alternate scan schedule for the Halo XR+ lidar. The observations are typical for bores where there are strong initial waves followed by elevated turbulence along with a lifting of the boundary layer (Haghi et al., 2019; Haghi and Durran, 2021), which is discussed in more detail in the

following sections. It is also important to note that the gravity waves manifest most strongly in the wind speed signal at heights from 200-800 m a.g.l. The intensity of the gravity waves in the wind speed signal diminishes closer to the surface and towards the top of the rotor layer, which was also found to be true in the radar measurements (Fig. 3).

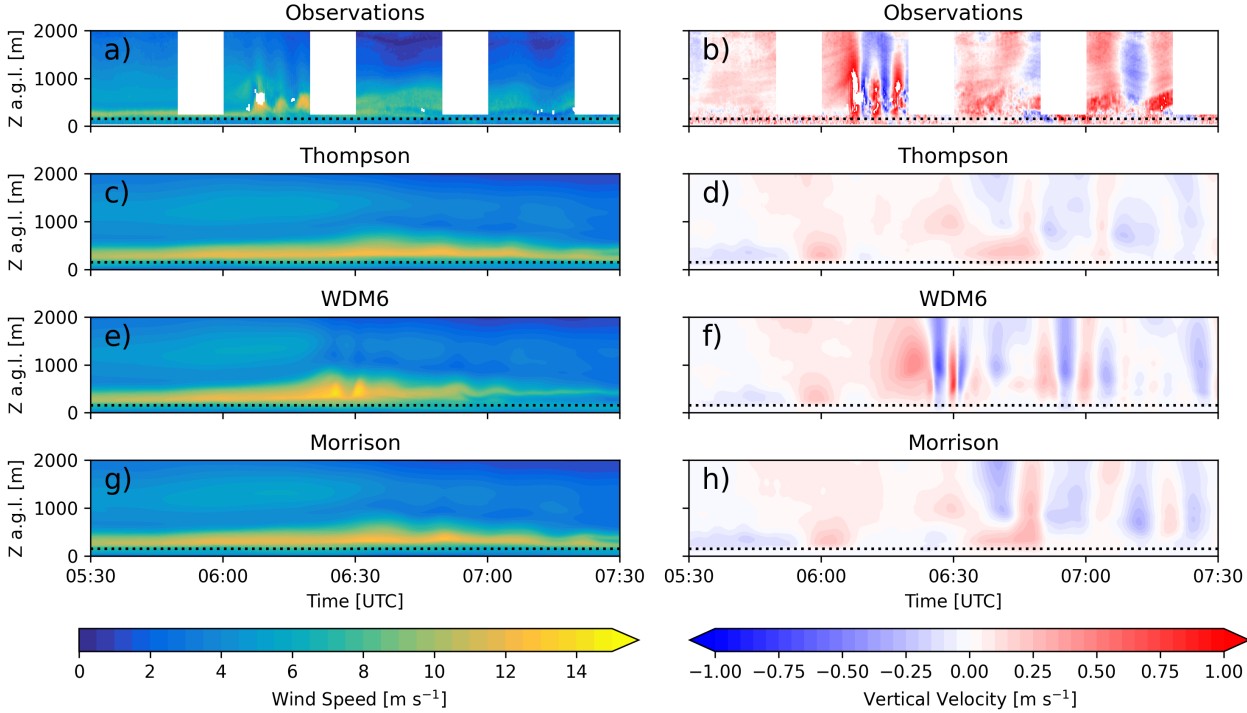

**Figure 7.** Time-height contours of wind speed and vertical velocity in (a,b) observations from the lidars and (c-h) the model for various microphysics schemes all at the Site A1 location. The dotted line corresponds to the top of the wind turbine rotor layer.

## 5.2 Gravity wave characteristics and evolution

The MCS-induced cold pool propagates outward, which perturbs the stably stratified atmosphere as a bore. The progression of a bore is well-described in Haghi et al. (2017) with the bore in the present study falling between a bore with undulations and one with solitary-like waves by the time the bore reaches the AWAKEN site. Note that all results presented in this manuscript hereinafter use the WDM6 microphysics scheme. Figure 8 shows instantaneous plan slices at 30 minute increments for the vertical velocity at 1 km a.g.l. and the potential temperature at 200 m a.g.l on domain d01. The height of 1 km a.g.l. is a

standard for studies involving bores while 200 m a.g.l. represents a height closer to the surface and also where the cold pool is very strong. At 05:00 UTC, the bore forms a ring as it propagates outward and spreads radially from the parent MCS. As the bore moves closer to the AWAKEN region, it gradually mixes into the ambient environment. While the strength of the cold pool weakens the further it travels from its parent MCS, gravity waves continue to propagate in the thermally stratified atmosphere

as evident in Fig. 8. At 05:00 UTC, there is a clear leading wave, and by 05:30 UTC, trailing waves are seen forming a wave train. The wave train reaches the AWAKEN region at 06:30 UTC with two large waves followed by a number of additional waves in the train. The waves then weaken in intensity at 07:00 UTC.

The vertical structure of the atmosphere is substantially impacted by the propagating bore and associated wave train. Figure 9 shows a vertical cross-section, along the transect illustrated in Fig. 5(a), of the wind speed and potential temperature at the same time increments as in Fig. 8. Note that the colorbar in Fig. 9 is specifically chosen to highlight the colder temperatures which form the SBL (but note that the entirety of the atmosphere in Fig. 9 is stably stratified). At 05:00 UTC (Figs. 9(a) and 9(b)), the bore is evident as an injection of faster wind speeds and colder temperatures into the transect with the density current traveling in the positive direction. Initially, the colder temperatures in the bore appear most clearly as a singular front that perturbs the SBL. Over time, the front evolves into a number of wavy structures at the interface of the SBL and free atmosphere that form the wave train. Additionally, the bore itself increases the height of the SBL, which is discussed in more detail in Sect. 5.4.

In the wind speed signal, the interaction between the bore and ambient environment is more complex. At 05:00 UTC (Fig. 9(a)) and upwind of the density current (distances $> 100$ km) there is a pre-existing east-northeasterly LLJ with a height of 600 m a.s.l. and a maximum wind speed of 10 m s$^{-1}$. Also evident in Fig. 9(a) is the wind speed behind the bore front, which is greater than 14 m s$^{-1}$. Over time, the wind speeds within the bore weaken substantially from as high as 14 m s$^{-1}$ to 6-8 m s$^{-1}$. However, the bore continues to propagate in the positive direction, which results in the advection of the pre-existing LLJ in the positive direction. By 06:30 UTC (Fig. 9(g)), the effect of the bore on the preexisting LLJ manifests itself as two well-defined waves in the wind speed signal (with a number of additional wavy structures behind the front). These two maxima result in the clear, distinct waves in the time-height contours at A1 (Fig. 7(e)) that occurred 5 minutes after each other. The distance between the well-defined maxima is 4.5 km. Trailing waves are also evident in Fig. 9(g) and 9(h) but are much slower in wind speed and also have larger distances between the waves. By 07:00 UTC (Fig. 9(i)), the effect of the bore is that the LLJ no longer extends into the AWAKEN region and, as a result, the wind turbines at King Plains experience much slower wind speeds. The effect of this modulation of the mesoscale environment by the bore is further quantified in Sect. 5.4.

### 5.3 Bore and gravity wave effects on the boundary layer

The gravity waves associated with the bore are strongest higher in the atmosphere. Vertical slices of wind speed and vertical velocity for domain d02 are shown at various times during the gravity wave passage in Fig. 10. The vertical slices are taken along a north-south transect that intersect point A1 shown in Fig. 4. This transect intersects four rows of the King Plains wind farm but since the hub-height winds are more northeasterly, wakes are largely out-of-plane in Fig. 10. At 06:25 UTC, the first wave of the solitary-like wave train, which has a wave-to-wave time-interval of 5 minutes and physical distance of 4.5 km, is seen approaching the wind farm in Figs. 10(a) and 10(d). As time progresses, there are a number of additional waves that pass over the wind farm. While the gravity waves are the dominant feature above the wind turbines, there are other turbulent features related to wakes and ambient turbulence in the rotor layer and closer to the surface.

Vertical profiles provide a more quantitative understanding of how the structure of the atmosphere is modulated by gravity wave-wind farm interactions. Figure 11 shows vertical profiles of time-averaged wind speed, wind direction, potential tempera-

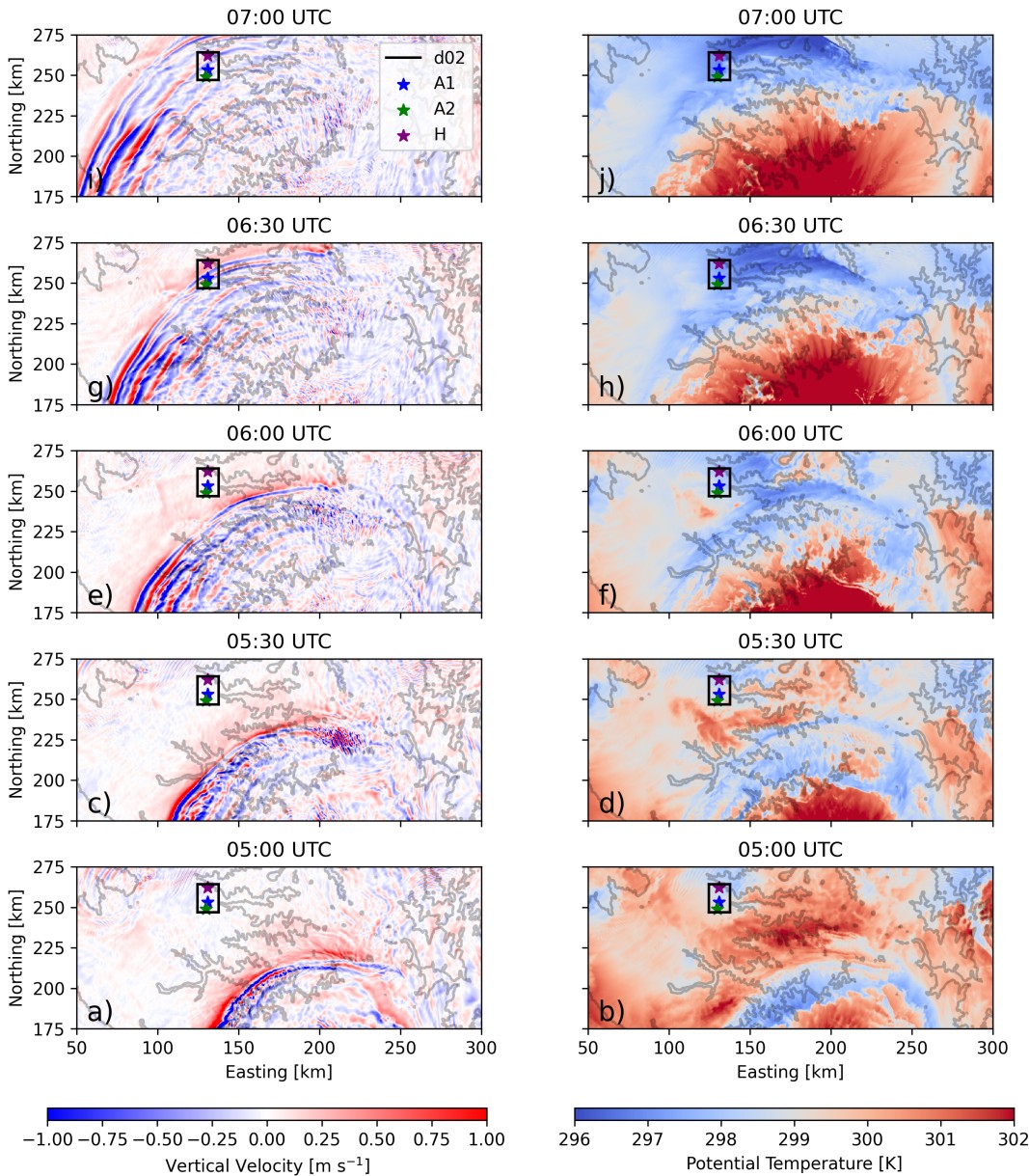

**Figure 8.** Plan slice of (a-e) vertical velocity at 1 km a.g.l. and (f-j) potential temperature at 200 m a.g.l. from 05:00 to 07:00 UTC in 30 minute increments on domain d01. Line contours represent 100 m changes in elevation. Note that the domain has been cropped for visualization purposes, and that the panels progress in time from bottom to top to highlight the northward movement of the bore. An animation of modeling results are included in the supplementary material (see Video 3, in the Video Supplement).

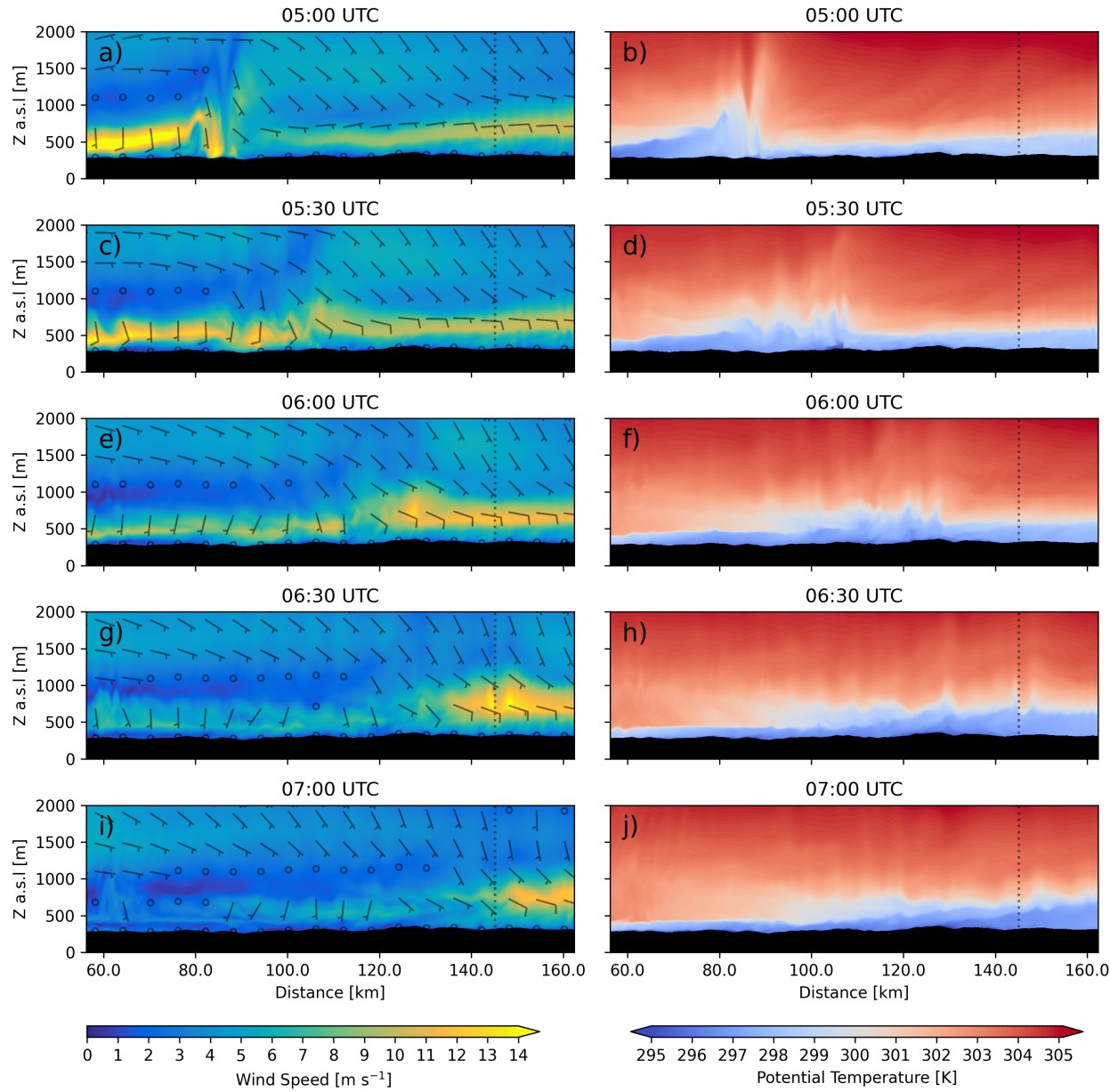

**Figure 9.** Vertical cross-section of wind speed (with wind barbs) and potential temperature from 05:00 to 07:00 UTC in 30 minute increments. The transects are along the dashed line shown in Fig. 5. The dotted line corresponds to the location for Site A1.

ture, and resolved heat flux at Site A1. At Site A1, model results are compared against both profiling (closer to the surface) and scanning (higher up in the atmosphere) Doppler lidar measurements of the wind speed. The vertical profiles are time-averaged

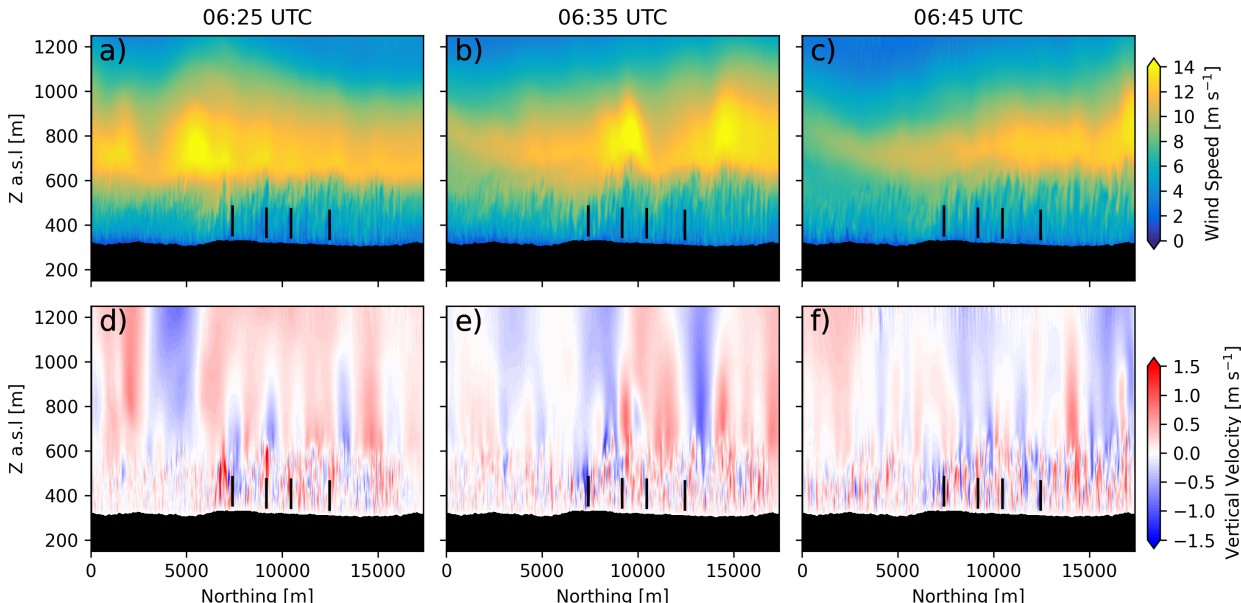

**Figure 10.** Instantaneous vertical cross-sections of (a-c) wind speed and (d-f) vertical velocity along the A1 north-south transect at various times during the gravity wave event.

**Table 3.** Time windows used for averaging vertical profiling results when comparing WRF-LES-GAD with observations at Site A1.

|  | Before | During | After |
|---|---|---|---|
| Measurements Averaging Window (UTC) | 05:34-06:04 | 06:04-06:34 | 06:34-07:04 |
| Model Averaging Window (UTC) | 06:00-06:20 | 06:20-06:50 | 06:50-07:20 |

for periods before, during, and after the gravity wave passage. The averaging windows used depend on the specific site due to the time it takes for the gravity waves to propagate northward. We assume that the gravity waves propagate at a speed equal to the wave-to-wave physical distance (4.5 km) divided by the time-interval (5 minutes), which is 15 m s$^{-1}$. Additionally, because there is some difference in when the wave arrives in the model compared to the measurements, there is a 16 minute shift used when defining these periods for the measurements compared to the model (as was done when similarly analyzing a MCS by Tomaszewski and Lundquist (2021)). For example, the averaging window used for "during" is 06:04-06:34 UTC at Site A1 for the measurements, while it is 06:20-06:50 UTC for the model. The averaging periods used to generate the vertical profiles are summarized in Table 3.

The measurements and modeling results demonstrate the weakening and eventual destruction of the LLJ due to the propagating gravity waves generated by the MCS. In the vertical profiles of wind speed (Fig. 11(a)), there is a well-defined maximum

above the rotor layer (representing the LLJ nose) prior to the gravity waves. During the gravity wave passage, the vertical location of the LLJ nose increases in height and after the gravity waves, the LLJ maximum is significantly diminished. Although, the wind speed maximum of the LLJ and the negative shear region above are both overestimated in the model. However, in both observations and the model, the LLJ itself does not recover beyond the "after" analysis period prior to sunrise (not shown). In addition to the decrease of wind speed due to the gravity waves, the wind direction also becomes more easterly as shown in

Fig. 11(b). Before the gravity waves arrive, the wind direction is east-northeasterly (approximately 60 degrees) at hub-height, with substantial wind veer across the rotor layer. During and after the gravity wave passage, the wind direction is more easterly, which is suboptimal for the east-west rows of the King Plains wind farm, due to waking within the row. The bore and the associated gravity waves advect from the south-southeast, which forces the ambient northeasterly winds to become more easterly. The effect of the gravity waves on the wind direction is discussed further below.

The thermal structure of the atmosphere is also strongly impacted by the gravity wave passage. Prior to the gravity wave arrival, the boundary layer is stably stratified up to approximately 200 m, above which the atmosphere is more stable. In Fig. 11(c), the boundary layer height is increased during the gravity wave passage. The increased boundary layer height persists even after the gravity wave passage, which is a common characteristic of bore events (Haghi et al., 2019; Haghi and Durran, 2021). In addition to the boundary layer height increasing, the boundary layer cools and the thermal stratification

weakens during the gravity wave passage, and these changes also persist after the wave passage. The stability weakening was also observed by the sonic anemometers at the A1 surface met station as previously discussed in Sect. 2.1. The cooling and deepening of the boundary layer are driven in part by the advection of colder temperatures into the region by the bore itself. Additionally, as the gravity waves pass, there is a positive heat flux in the upper atmosphere (Fig. 11(d)), indicating that heat is transferred from lower altitudes to higher altitudes, thus weakening the thermal stratification.

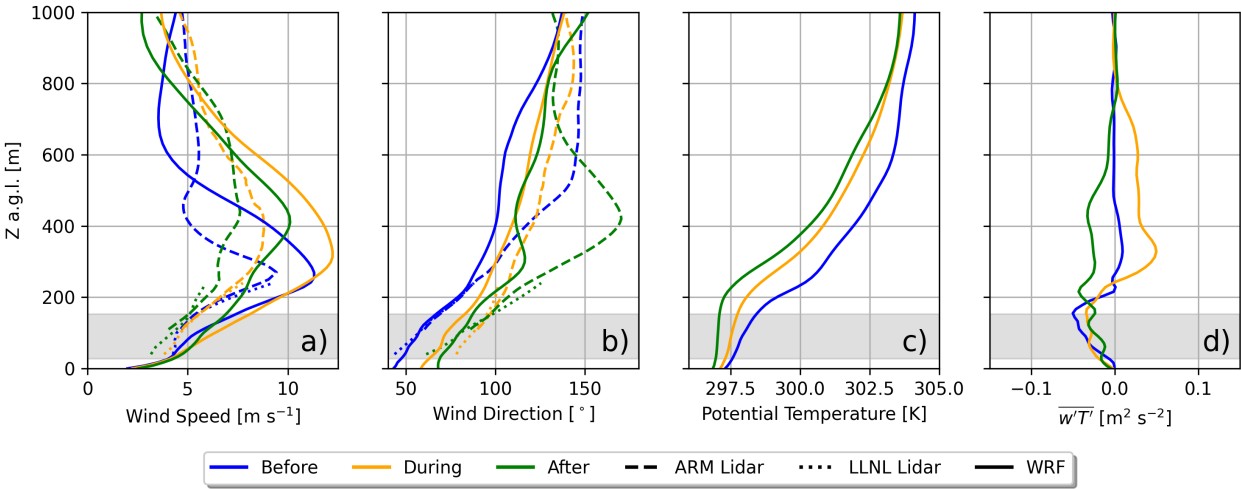

**Figure 11.** Time-averaged vertical profiles of (a) wind speed, (b) wind direction, (c) potential temperature, and (d) resolved heat flux for observations and simulation results at Site A1. See Table 3 for the averaging windows. The wind turbine rotor layer is highlighted in gray.

## 5.4    Gravity wave - wind farm interaction

The effect of the gravity waves on the power production of the King Plains wind farm is two-fold. First, the destruction of the LLJ by the gravity waves results in a decrease in the average power production. Second, there is increased power variability during the gravity wave passage. Figure 12(a) shows a time-series of the simulated power production of the wind farm by row, with the power signal smoothed to reduce noise by applying a 1-minute running average. Figure 12(b) shows a time-series of the observed power production for row 1 from the SCADA data, with the power data normalized at the request of the wind farm operator. The King Plains wind farm layout is non-uniform and as a result, there are certain rows with more wind turbines than others. In an attempt to equalize this factor, three turbines in each row are selected as shown in Fig. 4. Additionally, because the gravity wave propagation direction and turbine rows are slightly misaligned, only the three middle turbines from each row are considered to quantify the coherent effect of the gravity waves on individual turbines. In Fig. 12, the period during which the turbines are most significantly impacted by the gravity waves is highlighted in gray (06:25-06:55 UTC). For the simulated power, prior to the gravity waves, the first row of turbines averages 1.54 MW of power, whereas after the gravity waves have modulated the mesoscale environment, the first row of turbines averages 44% less power at 0.86 MW. All rows of turbines produce less power after the gravity waves pass, with row 2 producing 56% less power, row 3 producing 34% less power, and row 4 producing 15% less power on average. Row 1 has the largest increase in power variability during the gravity wave passage. The standard deviation increases by over 100% for row 1 from 0.18 MW prior and 0.16 MW after the gravity waves to 0.42 MW during the gravity wave passage. The power variability in the other rows is not as notable.

The SCADA data qualitatively confirms many of the trends in the simulated power production in Fig. 12. The power output of all rows decreases after the gravity wave passage. The SCADA data similarly shows power variability due to the gravity waves (in the gray region). However, there are differences in the variability on timescales shorter than that of the gravity waves, due to differences between the actual and simulated turbine controller. The simulations and SCADA data both show clear effects of the gravity waves in the power signal for row 1, but for rows 2-4 (especially rows 3 and 4), the gravity wave effects are more extreme in the SCADA compared to the simulation results. A potential reason for this difference is that the observed gravity waves could contain more energy than those predicted by the model. In the model, the gravity waves near the surface are dissipated as soon as they encounter the first row of the wind farm. In reality, the more energetic gravity waves are likely able to entrain momentum from above such that their effect is felt more strongly throughout all four rows. Considering the good agreement between the simulated and observed power signals for row 1, the model is able to capture the leading edge of the gravity wave passage but likely overestimates the dissipative effect of the wind farm on the gravity waves. Lastly, it is important to note that subtle differences in the wind direction can cause large power fluctuations due to waking because of the configuration of the farm (as discussed in the following paragraphs).

The increased power variability during the gravity wave passage is due to oscillations in the wind speed magnitude (mainly for row 1) as well as due to an indirect effect, related to subtle shifts in the wind direction caused by the wave motion. Figure 13 shows the simulated power signal in rows 1 and 2 along with the local perturbation pressure and wind direction signals. The perturbation pressure represents the total atmospheric pressure with the base pressure removed (Skamarock et al.,

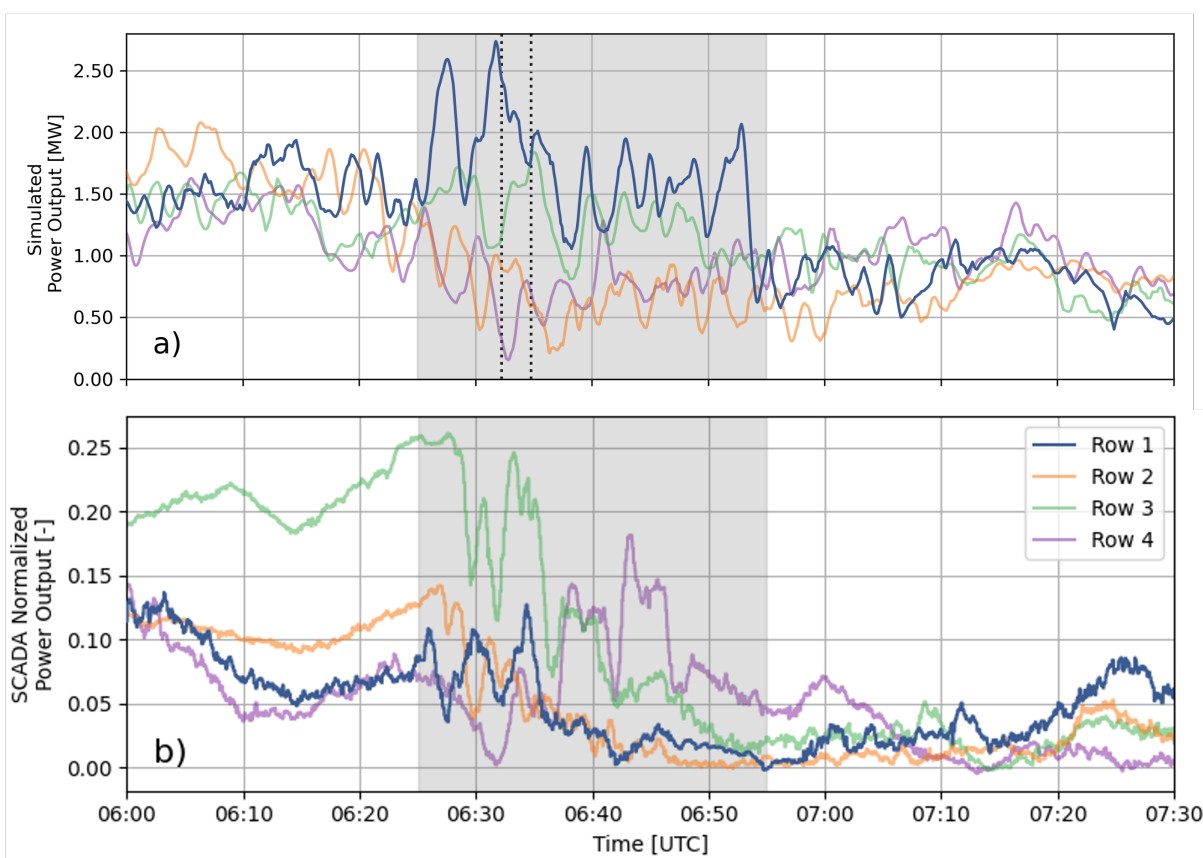

**Figure 12.** (a) Time-series of simulated row power outputs for the eastern King Plains wind farm from 06:00 to 07:30 UTC. The dotted lines in (a) highlight the time instants shown in Fig. 14. (b) Time-series of SCADA normalized row power outputs. Because of the delay in when the gravity waves arrive in the model compared to observations, the SCADA data has been shifted by 16 minutes. Figure 4 shows the turbines that represent each row.

2021). The local hub-height perturbation pressure and wind direction signals are obtained by spatially averaging a 600 m ×
600 m area centered over the middle turbine within each row. At the beginning of the period affected by the gravity waves, there are two peaks in the pressure signal which correlate to the waves in Fig. 7. The wind direction also correlates with the perturbation pressure signal, with higher perturbation pressure corresponding to more easterly wind and lower perturbation pressure corresponding to more east-northeasterly winds.

The correlation of the hub-height wind direction and perturbation pressure is qualitatively more clear in a plan view. Fig-
ure 14 shows the wind speed and perturbation pressure at hub-height along with streamlines at two different time instances. Closer to the surface, the gravity waves manifest most clearly in the perturbation pressure signal. The two time instances were chosen to display the effect of high and low pressure regions beneath the gravity waves. During the first time-instance (06:32:15 UTC), the second row of turbines is in a low pressure region and is experiencing wind directions that are more

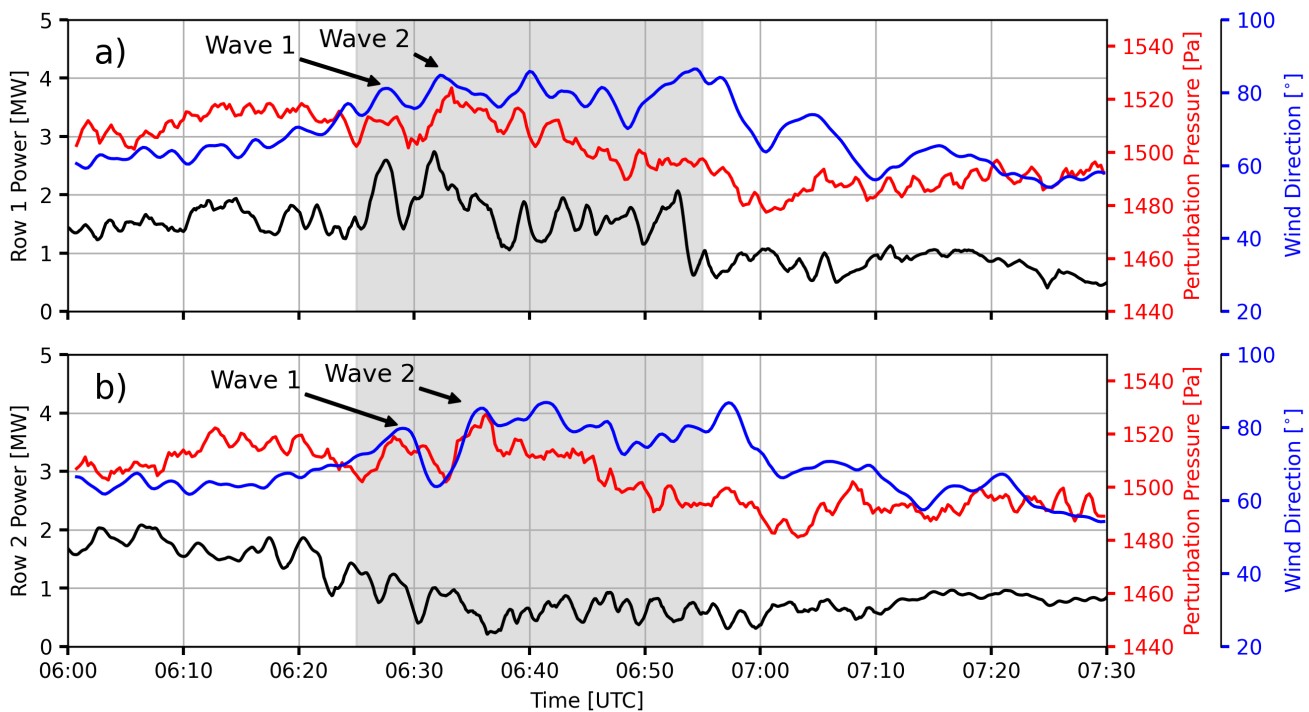

**Figure 13.** Time-series of the simulated (a) Row 1 and (b) Row 2 local wind direction, perturbation pressure, and power for the eastern King Plains wind farm from 06:00 to 07:30 UTC.

east-northeasterly. During the second time-instance (06:34:45 UTC), the second row of turbines is in a high pressure region
and the wind direction is easterly. The same relationship between pressure and wind direction is observed beneath the gravity
waves. More easterly winds are less favorable for the King Plains layout and result in significant power losses due to wakes.

The second row of turbines produces more power during the first time-instance with the east-northeasterly winds at 0.90 MW
while the same row produces 34% less power during the second time instance with easterly winds. A similar, but opposite,
trend happens for row 4 related to the wave-to-wave distance of the waves. During the first time-instance, the majority of the
450 turbines in row 4 are waked during the easterly wind directions and the row is producing just 0.32 MW of power. The turbines
in row 4 experience more east-northeasterly wind directions during the second time-instance and are thus producing more
power at 0.71 MW.

While the gravity waves induce wind direction shifts which modulate the produced power, it is important to remark that the
power production in the farm is variable due to additional factors. Importantly, local turbulent structures have a strong impact
on the wind speed experienced by turbines that increase or decrease power. Additionally, these local turbulent structures induce
wake meandering and, given the non-optimal wind directions for the layout, can increase or decrease turbine power production
depending on whether they are being waked or not.

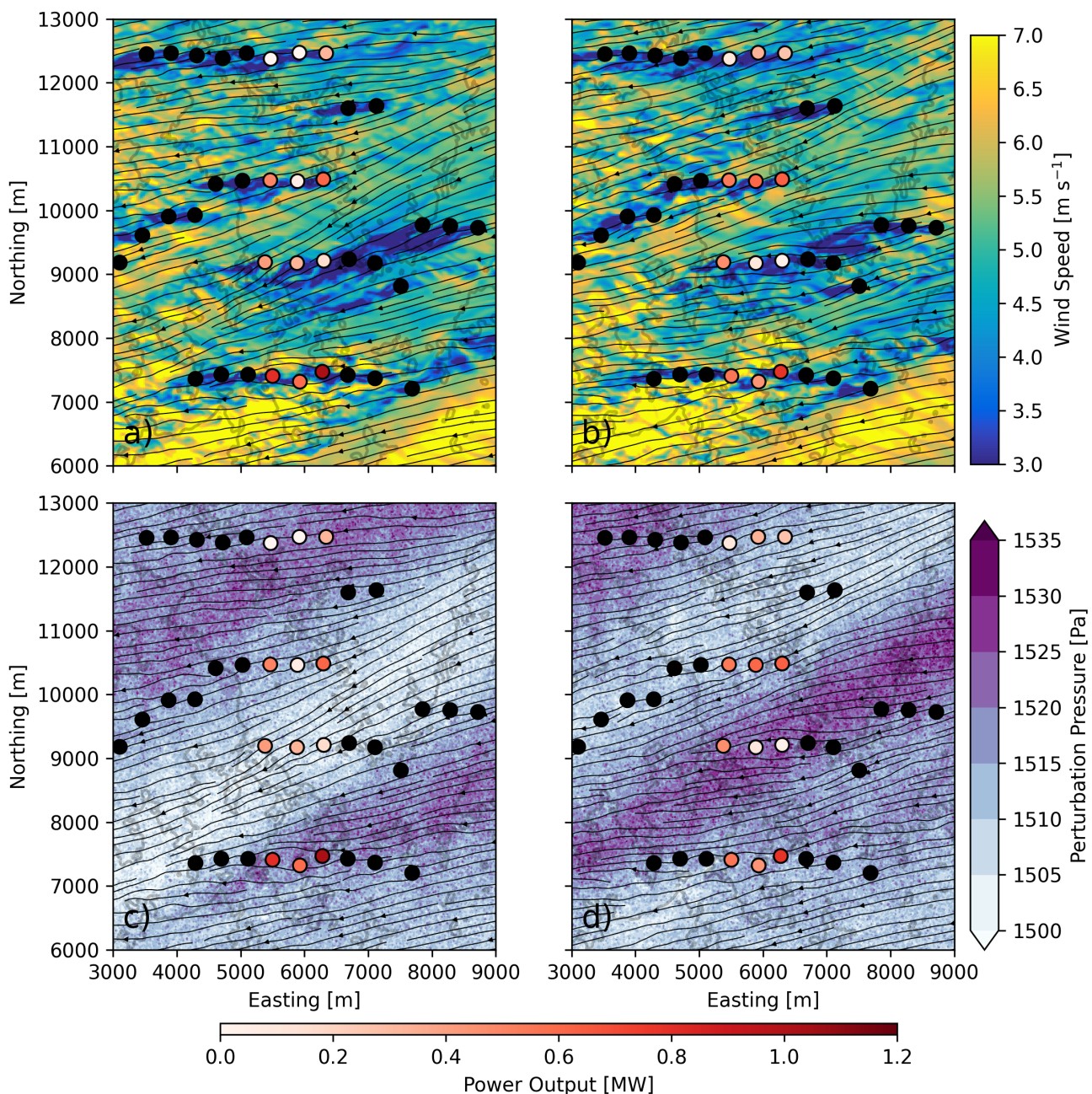

**Figure 14.** Instantaneous plan view of hub-height wind speed and perturbation pressure at (a,c) 06:32:15 UTC and (b,d) 06:34:45 UTC. The streamlines correspond to the hub-height wind speed; however, note that the gravity waves are advecting northward. An animation of modeling results are included in the supplementary material (see Video 4, in the Video Supplement).

## 6 Conclusions

In this study, WRF-LES-GAD is used to understand wind farm impacts from gravity waves observed on 06 June 2023 during the American Wake Experiment (AWAKEN) field campaign. The case study was initially characterized using X-Band and NEXRAD weather radars as these devices measure very large geographical extents. The gravity waves were determined to be associated with an atmospheric bore event, which was generated by nocturnal mesoscale convection near the AWAKEN site. A cold pool, which formed as a result of precipitation-induced latent cooling within the mesoscale convective system (MCS), began to spread radially as a density current or bore. A train of solitary-like waves formed at the front of the bore, interacting with the preexisting stable boundary layer in the study region and ultimately passing through the King Plains wind farm.

To model the bore and its associated gravity waves, we use a two-domain nested setup in WRF-LES-GAD with grid spacings of 300 m and 20 m. Both domains use an LES closure. The nested grids are forced by the 3 km HRRRv4 model, which best captures the MCS by assimilating NEXRAD data. In this setup, the 300 m domain serves as the MCS-resolving grid, while the 20 m domain resolves ambient, small-scale turbulence as well as wake-generated turbulence. The effect of 50 wind turbines representing the eastern half of the King Plains wind farm are included on the 20 m domain using a generalized actuator disk approach.

The bore formation, propagation, and ultimately the structure of the gravity waves traveling through the AWAKEN region was found to be highly sensitive to the microphysics parameterization, as found in previous work (Johnson et al., 2018). We investigated three different microphysics schemes: the Thompson parameterization (1.5-moment), the WRF 6-class (WDM6) parameterization (double-moment), and the Morrison parameterization (double-moment). Simulations with the WDM6 parameterization provided results with waves separated by 5 minutes, which best matched observations from lidars. The WDM6 scheme produces a more realistic bore structure because the rain water mixing ratio profile shows a strong decrease below the cloud base, compared to relatively constant profiles for Thompson and Morrison. The decrease in rain water mixing ratio results in more latent cooling and thus a stronger cold pool and density current. Additionally, the Thompson and Morrison parameterizations result in more snow and less graupel and rain at higher altitudes. These results align with those of Johnson et al. (2018) who hypothesized that the increase in snow hydrometeors results in lower fall velocities in comparison to rain drops and thus less precipitation.

For the fine-scale domain with 20 m horizontal grid spacing and the wind turbines parameterized, the ambient turbulence was sensitive to the LES closure. Considering that 20 m is relatively coarse for a SBL simulation, resolving ambient turbulence is a challenge for conventional closure schemes. While using finer grid resolution would be desirable, considering the large geographic extent of the region, 20 m grid spacing represented a compromise to reduce computational cost while still providing acceptable modeling results. For the given setup and case study, results with dynamic turbulence closures are more effective at capturing small-scale turbulence. This agrees with previous work that has shown that dynamic turbulence closures are more effective in SBL conditions and at coarser grid resolutions compared to conventional closures (Zhou and Chow, 2011, 2012, 2014; Wise et al., 2024).

The bore passage and associated gravity waves generated by the MCS affect both the average power production and the power variability during the study period. Prior to the bore event, the wind turbines experienced an easterly low-level jet (LLJ), but the passage of the bore and gravity waves modulated the mesoscale environment substantially. After the gravity wave passage, the LLJ is effectively destroyed, with a much weaker jet nose resulting in reduced hub-height wind speeds, as was seen in both observations and modeling results. The LLJ is weakened because the gravity waves increase the SBL height, which is a common characteristic of bore events. As a result, the average power production decreases after the gravity wave passes, by up to 56% depending on the turbine row and 39% on average for all four rows. During the gravity wave passage, the power variability is increased by over 100% for the most southern row of turbines. Similar trends were observed in wind turbine SCADA data. The increase in power variability is shown to result from wind direction oscillations associated with the gravity wave passage with wind variability related to wave-induced turbulence also playing a role. The easterly wind direction is suboptimal for the King Plains wind farm since the east-west layout was designed for a predominantly southerly wind direction.

The WRF-LES-GAD modeling setup presented here provides a framework for simulating atmospheric bore and gravity wave effects on a wind farm. This case study shows good agreement between model results and available observations, demonstrating the efficacy of this approach for understanding the interaction between realistic gravity waves and wind farms. Gravity waves are a common feature in the Southern Great Plains, coming from many directions and with various amplitudes and periods. The modeling framework used here is well-suited for other gravity wave case studies from the AWAKEN field campaign. Future work will focus on characterizing the different types of gravity wave events observed during AWAKEN, to better understand their formation mechanisms and their effects on wind farm power production.

*Data availability.* The full WRF-LES-GAD simulation data is several terabytes but subsets of the data can be shared upon request. The source code can be found at https://github.com/adamwise95/WRFv4.4-DRM_GAD

*Video supplement.* The following is available online at https://doi.org/10.5281/zenodo.12551368, Video 1: NEXRAD WSR-88D radar reflectivity over central Oklahoma from 03:30 to 06:00 UTC on 06 June 2023. Video 2: Wind speeds from TTU X-Band radars at heights of 95, 145, and 270 m a.g.l. from 05:30-07:30 UTC. Video 3: Simulated vertical velocity at 1 km a.g.l. and potential temperature at 200 m a.g.l. on domain d01 ($\Delta x = 300$ m). Video 4: Simulated hub-height wind speed, perturbation pressure, and turbine power output on domain d02 ($\Delta x = 20$ m).

## Appendix A: Ambient turbulence sensitivity to closure scheme

For the AWAKEN model setup in the region surrounding the King Plains wind farm, domain d01 resolves the mesoscale bore and gravity waves, and the nested domain d02 additionally resolves smaller-scale ambient turbulence. At 20 m grid spacing and in stably stratified conditions, the role of the turbulence closure in representing ambient turbulence is critical, especially to accurately represent the dissipation of individual wind turbine wakes. While results using the DRM closure are used in the main text of this study, domain d02 is also run with the turbulent kinetic energy 1.5-order model (TKE-1.5) (Deardorff, 1980) and the Dynamic Wong-Lilly (DWL) model (Wong and Lilly, 1994) for comparison. The TKE-1.5 model (Deardorff, 1980) is one of the standard turbulence closures used in WRF (and many other LES modeling tools). This model solves a prognostic turbulent kinetic energy (TKE) equation, which describes the evolution of TKE and parameterizes sources and sinks from shear production, buoyancy production or suppression, turbulent mixing, and dissipation. The Dynamic Wong-Lilly (DWL) model is the subgrid scale component of the DRM model (the DRM model uses a combination of the DWL eddy-viscosity model and and the scale-similar RSFS term to create a mixed-model for turbulence) with the DWL model using the explicit filter to dynamically solve for the eddy viscosity coefficients of interest (Lilly, 1992).

Figure A1 shows an instantaneous plan slice of hub-height wind speed at 06:00 UTC (prior to the gravity wave and after 1 h of spin up). Qualitatively, the wind speeds are very similar across all three closures; however, the fine-scale structure of turbulence is different. The wind direction is east-northeasterly and wakes are seen as the slower wind speeds downwind of each individual turbine. For the TKE-1.5 closure, there are patches in the flow where there is very little turbulence, especially in the inflow regions upwind of the farm. Figure A2 highlights an inflow region in Fig. A1, with Fig. A2 showing both the wind speed and the vertical velocity where the qualitative differences in the wind speed and vertical velocity variance are more clear between the TKE-1.5 closure and the dynamic closures. In general, for the DWL and DRM closures, the small-scale wind speed variability is more uniform, which is more realistic for ambient turbulence.

High quality hub-height turbulence observations are limited in the AWAKEN region; however, wind speeds in the rotor layer were measured at relatively high temporal frequency (∼0.25 Hz) by the profiling Doppler lidar at A1. The lidar at A1 was a pulsed lidar, which allowed for this faster scan frequency. Using these measurements, the power spectra for the hub-height wind speed can be generated in time from 06:00 to 06:25 UTC (prior to the gravity waves) shown in Fig. A3, which can be directly compared with simulation results at A1. Note that the 16 minute delay in which the gravity waves arrive in the model has been accounted for with the time period for the lidar in Fig. A3(a) and A3(b) being shifted to 05:44 - 06:09 UTC. The spectra are computed using Welch's method (Welch, 1967). Overall, there is reasonable agreement between the modeling results and measurements with limitations related to the relatively coarse grid resolution. The modeling results and measurements agree well in the inertial subrange and follow a slope of -5/3 for part of the spectrum (Kolmogorov, 1941); however, the larger-scale or lower-frequency structures are overestimated and the smaller-scale or higher-frequency structures are underestimated by the model.

The dynamic closures contain more energy at higher frequencies compared to TKE-1.5, as expected, with the DRM simulation containing the most energy in a frequency range of 0.012 - 0.016 Hz (∼60-80 s) compared to the other closures. At

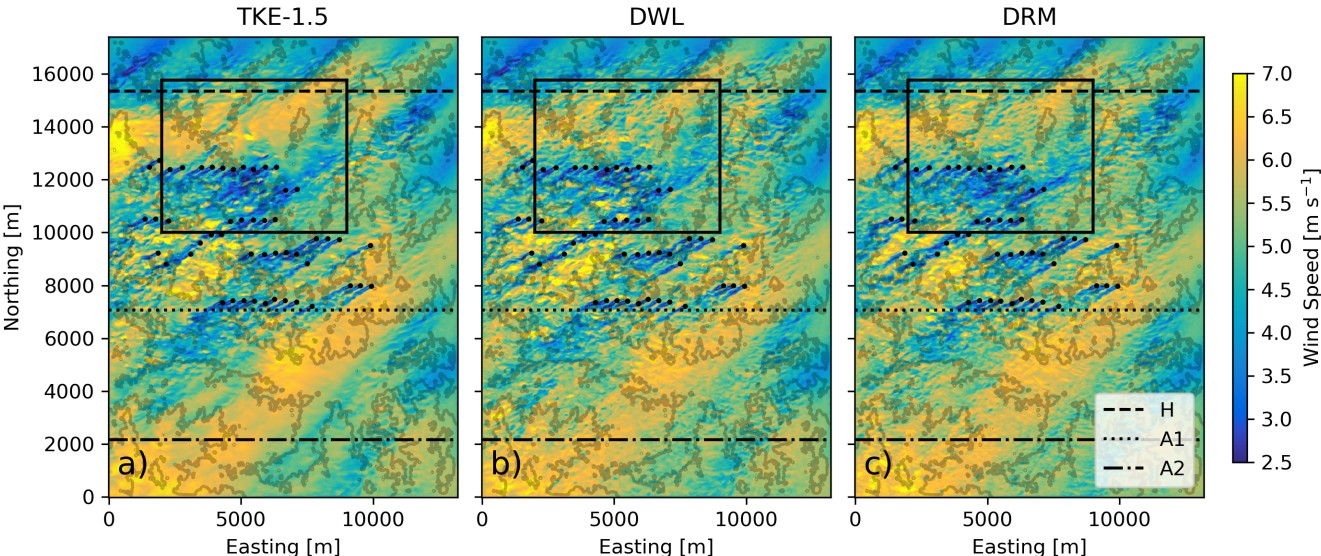

**Figure A1.** Plan slice of hub-height wind speed at 06:00 UTC in domain d02 for simulations results using the (a) TKE-1.5, (b) DWL, and (c) DRM turbulence closures. The area outlined in black is the extent shown in Fig. A2. Line contours represent 15 m changes in elevation.

higher frequencies (above ~0.02 Hz), the reduction in energy due to the effect of the grid-cutoff is apparent, as is typical for finite difference schemes Skamarock (2004). Additionally, because the high-frequency turbulence is underestimated, the energy content is shifted to lower frequencies, which results in an overestimation of larger-scale turbulence in the model compared to observations. A similar shift occurred in SBL conditions for Wise et al. (2022) and in neutral boundary layer conditions for Wiersema et al. (2022), and is indicative of large-scale structures not cascading into smaller-scale motions limited by the grid resolution. Importantly, the hub-height wind speed and turbulence intensity are well-predicted by the model during the analysis period. For all three closures, the hub-height wind speed biases are under 0.4 m s$^{-1}$ and the mean absolute errors are less than 1.0 m s$^{-1}$. For the DRM closure, which we conclude to perform best in the spectral analysis, the model predicted mean hub-height wind speed is 4.50 m s$^{-1}$ compared to an observed wind speed of 4.24 m s$^{-1}$ (the low values are because Site A1 is waked during the analysis period). Additionally, the mean model-predicted turbulence intensity is 17.9% compared to an observed 16.5% during 06:00 to 06:25 UTC.

To quantify the spatial variation in turbulence between the three closures, we use energy spectra in the vertical velocity signal along various east-west transects in domain d02. Figure A4 shows spectra in wave-space of the vertical velocity for the three different closures taken along east-west transects that intersect Sites H, A1, and A2. Hereinafter, these transects will be denoted as Transect H, Transect A1, and Transect A2. The spectra are obtained following the methods of Durran et al. (2017) and Connolly et al. (2021) and consist of 650 points along the transect (excluding 5 points near the eastern and western boundaries). The spectra shown in Fig. A4 are an average of the individual spectra calculated at each output time step (every 15 s) from 06:00 UTC to 06:25 UTC, prior to the gravity wave interaction. Transect H can be considered as purely inflow, while

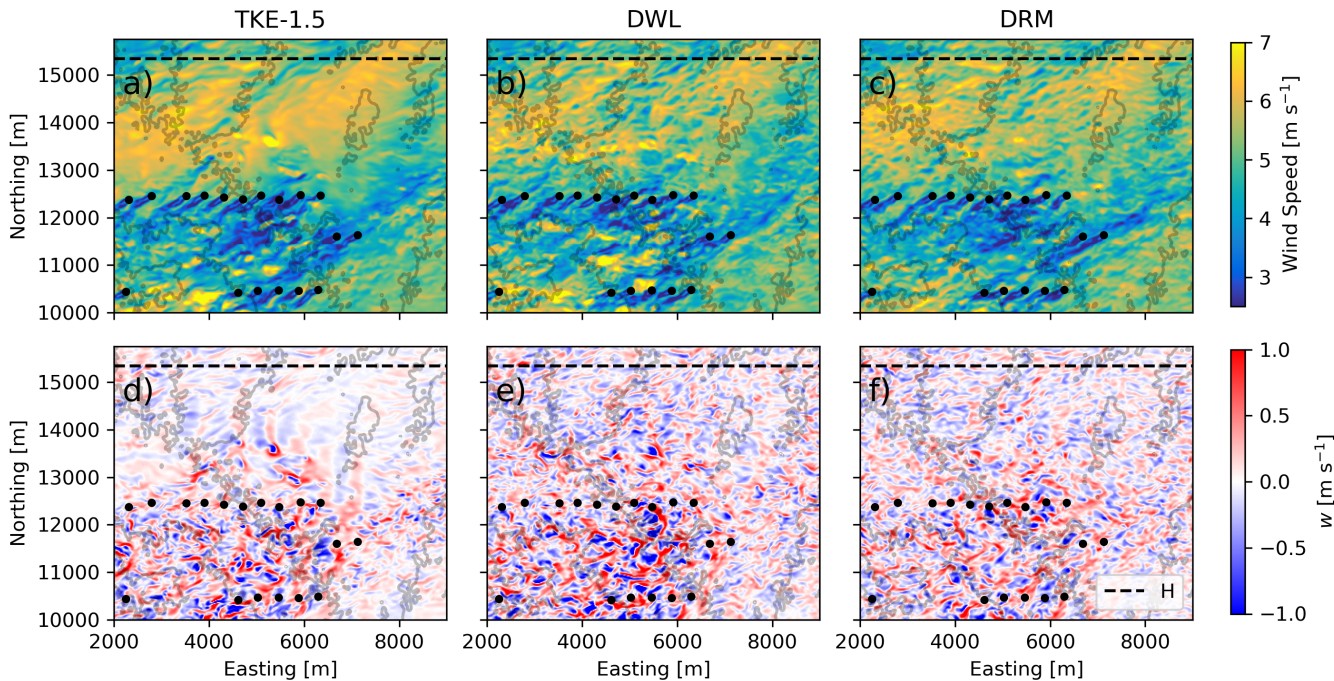

**Figure A2.** Zoomed-in plan slice of hub-height (a-c) wind speed and (d-f) vertical velocity at 06:00 UTC in domain d02 for simulations. Line contours represent 15 m changes in elevation.

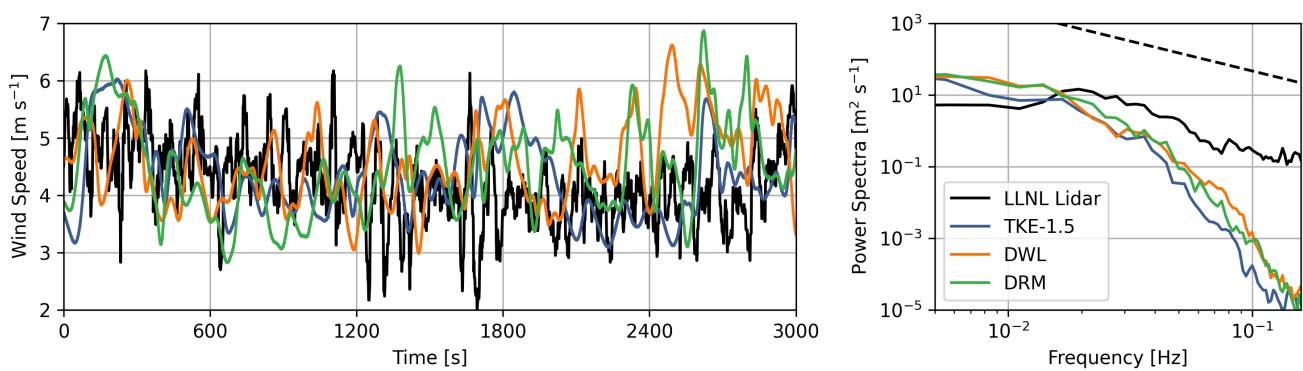

**Figure A3.** Time-series (a) and power spectral density (b) of wind speed at Site A1 for the observations and simulation results with the various closures prior to the gravity waves. The analysis period for the modeling results is 06:00 - 06:25 UTC and 05:44 - 06:09 UTC to account for the delay in which the gravity waves occur in the model. The dashed black line in (b) represents the -5/3 energy cascade range in the inertial subrange.

the flow at other transects is affected by wind turbine wakes. Along Transect H (Fig . A4(a)), there is increased energy for larger

wavenumbers (smaller wavelengths) when using the DWL and DRM closure compared to TKE-1.5, indicating that turbulence is more evenly distributed for the DWL and DRM closures. A similar trend holds true for Transects A1 and A2 (Figs. A4(b) and A4(c)) but to a much smaller degree. Additionally, transects A1 and A2 contain more energy in the smaller scales, as these locations are downstream of the wind farm and therefore contain added energy from wind turbine wake turbulence.

Ultimately, the DRM closure was used for further analysis as it best represents the ambient turbulent structures and provides
good agreement in the hub-height wind speed and turbulence intensity. There has been significant research on the sensitivity of LES modeling results in SBL conditions to the turbulence closure, and the present work agrees with previous findings in that dynamic turbulence models are able to resolve more turbulence in stably stratified conditions (Zhou and Chow, 2011, 2012, 2014; Wise et al., 2024). Interestingly, other simulations of the same site at the same grid resolution but focused on shear-driven instabilities in strong SBL conditions showed dramatic differences in the quality of ambient turbulence to the closure (See Appendix
E in Wise (2024)). In contrast with the study of Wise (2024), the current setup takes advantage of HRRRv4 forcing and the CPM, which is one reason why there are smaller differences in the representation of turbulence when using the different closures in this study. Another likely reason is that the strength of stratification is weaker in the present study compared to Wise (2024).

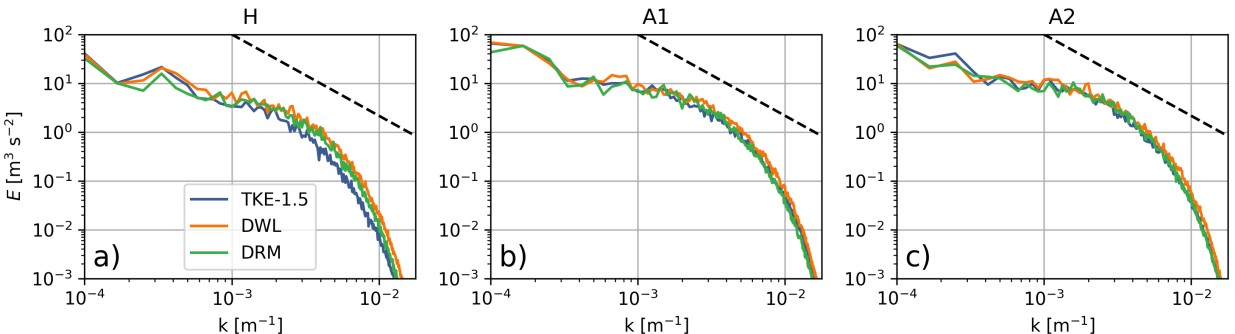

**Figure A4.** Wave-space spectra of vertical velocity for the various closures along the (a) H , (b) E, and (c) A1 east-west transects. Spectra are calculated every 15 s from 06:00 to 06:25 UTC and then averaged. The dashed black line represents the -5/3 energy cascade range in the inertial subrange.

*Author contributions.* Writing—original draft preparation and visualization, ASW; writing—review and editing, RSA, AA, SW, RK, RN,
BH, HS, PM, and FKC; methodology, software, validation, and formal analysis, ASW, RSA, AA, and FKC; conceptualization, ASW, RSA, PM, and FKC; investigation and data curation, AA, SW, RK, RN, BH, and JS. All authors contributed with critical feedback on this research and have read and agreed to the published version of the manuscript.

*Competing interests.* The authors declare that they have no conflict of interest.

*Acknowledgements.* Correspondence with Aaron Johnson of the University of Oklahoma regarding bore modeling in WRF was greatly ap-
preciated. This research was supported by the Wind Energy Technologies Office of the U.S. Department of Energy office of Energy Efficiency
and Renewable Energy through a PO with Sandia National Laboratories as part of the multilab AWAKEN project. This material is based
upon work by ASW supported by the National Science Foundation Graduate Research Fellowship Program under Grant No. DGE 1752814.
This research used the Savio computational cluster resource provided by the Berkeley Research Computing program at the University of Cal-
ifornia, Berkeley (supported by the UC Berkeley Chancellor, Vice Chancellor for Research, and Chief Information Officer). RSA's and SW's
contributions were prepared by Lawrence Livermore National Laboratory under Contract DE-AC52-07NA27344, with support from the U.S.
DOE Office of Energy Efficiency and Renewable Energy Wind Energy Technologies Office. PNNL is operated for DOE by the Battelle
Memorial Institute under Contract DE-AC05-76RLO1830. This work was authored [in part] by the National Renewable Energy Laboratory,
operated by Alliance for Sustainable Energy, LLC, for the U.S. Department of Energy (DOE) under Contract No. DE-AC36-08GO28308.
Funding provided by the U.S. Department of Energy Office of Energy Efficiency and Renewable Energy Wind Energy Technologies Of-
fice. The views expressed in the article do not necessarily represent the views of the DOE or the U.S. Government. The U.S. Government
retains and the publisher, by accepting the article for publication, acknowledges that the U.S. Government retains a nonexclusive, paid-up,
irrevocable, worldwide license to publish or reproduce the published form of this work, or allow others to do so, for U.S. Government
purposes.

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
