# Peer review of "Large-eddy simulation of an atmospheric bore and associated gravity wave effects on wind farm performance in the Southern Great Plains"

_Wind Energy Science, 2024_

## Author Comment (AC1)

**Reviewer 2**

The manuscript describes a case study of interactions between internal gravity waves (a bore initiated by nearby convection) and wind turbines in a wind farm, in Oklahoma. The case study uses a two-domain nested simulation, with the horizontal resolution down to 20m in the nested domain. Although the simulations present several difficulties (sensitivity of the convection and ensuing bore to the microphysics scheme, fine-scale processes, eg wakes, to include in the wind farm) the authors show convincing simulations, supported by the careful comparison to the observations. Publication after minor revisions is recommended.

We thank the reviewer for the time they spent reviewing our work and for their supportive suggestions and comments. Our replies are inserted below into the reviewer's comments.

Minor comments:

l34: there is a case study by Ralph, Neiman and Keller, 1999, worth mentioning with respect to fronts generating gravity waves near the surface

Thank you for bringing this study to our attention. It has now been included and cited in the introduction.

l39-49: repetition: '... in the US Great Plains. In the US Great Plains...'

The second "in the US Great Plains" has been removed.

l63: affects -> affect

Thank you, this has been corrected.

l124: it would be useful to indicate the local time already at ths stage of the manuscript; it is done later on of course, but the reader wonders when first encountering the time...

The local time is now stated at this point.

l146: the three numbers are odd, for the Obukhov length: the unit is not given, and two numbers are negative...

The text now includes the units, which are in meters. The negative Obukhov lengths represent unstable atmospheric conditions as the heat flux close to the surface is positive (upwards) in the following expression:

$$L = -\frac{u_*^3 \theta}{\kappa g \overline{w'\theta''}},$$

Where $u_*$ is the friction velocity, $\theta$ is the potential temperature, $\kappa$ is the von Karman constant, $g$ is the gravitational constant, and $w$ is the vertical velocity. $\overline{w'\theta'}$ represents the heat flux close to the surface. The fluxes are calculated using 30-minute averages to calculate the perturbations (line 117 in the manuscript).

l156-157: what is meant by 'or in the operational mode..'?

This sentence has been clarified to state that the operational mode is when turbines are operating with power-maximizing control:

"Additionally, the hub-height wind speeds are relatively low, on the order of 4-6 m/s such that the power output of the farm is highly susceptible to wind speed fluctuations in this range (in region 2 of the power curve or in the operational power-maximizing control mode of the wind turbines)."

l198-199: the formulation is a bit surprising: 'the setup is unique (...) only two domains.' This is presented rather as a strength of the study, but using only two domains implies a strong ratio between the resolutions used in one and the other domain, which is generally considered unfavorable...

We agree that using a large ratio between the resolutions would traditionally be considered as non-standard. A major strength of using this setup is that it is computationally efficient, which has now been stated in line 203:

"The setup is unique compared to other multi-scale WRF setups in that it uses only two domains, which makes it very computationally efficient."

The case study is unique in that there are two very clear dominant scales in the flow, which makes a two-domain setup feasible. The first are the scales related to the gravity waves themselves and the second are scales related to ambient small-scale turbulence in stable atmospheric conditions. Through sensitivity studies conducted both by us and Johnson et al. (2019), a horizontal grid spacing of 300 m resolves the relevant and important scales related to the gravity waves. The domain with 20 m grid spacing resolves the ambient small-scale turbulence (finer resolution would be desirable but we are limited by computational resources).

Domains between 300 m and 20 m grid spacings represent a gray zone in stable conditions where turbulence is either not resolved at all or not resolved accurately. It is possible that adding another domain could improve the results; however, this is not guaranteed due to the uncertainties of turbulence modeling in the gray zone (e.g., Chow et al. 2019). Given that our 20 m domain is large enough to include adequate fetch, while also including perturbations to encourage the development of turbulence, the benefit of an additional domain would likely be minimal, with a significant penalty in terms of computational cost.

Lastly, and most importantly, we achieved good results with this nesting configuration. The simulations results compared favorably with observations compared to other nesting configurations done in an initial sensitivity study.

l215: the formuation is ambiguous: "0 hour forecast (which includes the hour..." If data has been assimilated up to that point, it is odd to call this a forecast.. this should be reformulated

We agree that this formulation is unclear therefore it has been removed. The sentence at lines 218-219 in the revised manuscript now reads:

> "In this study, we use the HRRRv4 analysis product which includes assimilated NEXRAD data."

**figure 4: it is somewhat disturbing to use the same colormap for the two panels, but with different ranges**

The caption now explicitly states that the colormaps have different ranges due to the maximum and minimum elevations within each domain.

**figure 12: plural: 'power output[s]'**

This has been fixed.

**l455: repetition of 'results'; vary the choice of words: simulations? flow variables?**

This has been changed to "simulations".

**l463: 'For the 20 m domain... ' formulation can be improved...**

This beginning of this sentence now reads:

> "For the fine-scale domain with 20 m horizontal grid spacing and the wind turbines parameterized, …"

**figure 14: the power output in the circles (white to red colors) is only indicated for the three central turbines... why?**

This figure aims to show the coherent effect of the gravity waves. The three central turbines from each row are included because the aggregate effect of the waves in the power signal is still clear. When more turbines are included, the effect of the gravity waves are distorted because the waves are misaligned with the rows. A sentence has been added at line 407-409 in the revised manuscript for clarity:

> "Additionally, because the gravity wave propagation direction and turbine rows are slightly misaligned, only the three middle turbines from each row are considered to quantify the coherent effect of the gravity waves on individual turbines."

**l476: 'by up to 56 %': it is good to give the extreme value, but other statistics (mean) could also be meaningful to include**

The mean percent decrease in power for all four rows is now included in the conclusion (line 497 in the revised manuscript), which is 39%.

**References:**

Miguel Sanchez Gomez, Julie K. Lundquist, Jeffrey D. Mirocha, Robert S. Arthur, Domingo Muñoz-Esparza, Rachel Robey; Can lidars assess wind plant blockage in simple terrain? A WRF-LES study. *J. Renewable Sustainable Energy* 1 November 2022; 14 (6): 063303. https://doi.org/10.1063/5.0103668

Chow, F. K., Schär, C., Ban, N., Lundquist, K. A., Schlemmer, L., & Shi, X. (2019). Crossing Multiple Gray Zones in the Transition from Mesoscale to Microscale Simulation over Complex Terrain. *Atmosphere*, *10*(5), 274. https://doi.org/10.3390/atmos10050274

Johnson, A., and X. Wang, 2019: Multicase Assessment of the Impacts of Horizontal and Vertical Grid Spacing, and Turbulence Closure Model, on Subkilometer-Scale Simulations of Atmospheric Bores during PECAN. Mon. Wea. Rev., 147, 1533–1555, https://doi.org/10.1175/MWR-D-18-0322.1.

---

## Author Comment (AC2)

**Reviewer 3**

This paper reviews the effect of a thunderstorm-downdraft generated atmospheric bore and associated gravity wave on a downstream wind farm in central Oklahoma. The wind farm was located in the American Wake Experiment (AWAKEN) field campaign domain. Overall, the paper is well written, the methods cogently presented, and conclusions sufficiently supported by the analysis. It would be interesting to see what effects, if any, the wind farms had on the downstream propagation of the bore/gravity wave, but perhaps the topic for another paper. A brief discussion of the "forecastability" (short-term) of such events, particularly in the context of power production, would have been illuminating (although other PECAN/AWAKEN papers are addressing this?). Specific comments:

We thank the reviewer for the time they spent reviewing our work and for their supportive suggestions and comments.

The PECAN papers are much more focused on forecastability of bores and other nocturnal convection events as the goal of their field campaign was to improve forecasts for weather and climate models. Many of their studies dig into the necessary ingredients to initiate nocturnal convection (Weckworth et al. 2019, for example). The following sentence has been added in the introduction at lines 64-65 in the revised manuscript to point the reader to the relevant references for forecastability:

> "For additional discussion on nocturnal convection initiation and the forecastability of MCSs and bores, in general, see Weckwerth and Romatschke (2019) and Weckwerth et al. (2019)"

However, the PECAN papers do not specifically analyze wind turbine power forecasting as it relates to bores, which would be a good avenue for future work.

Tomaszewski and Lundquist 2021 studied how a thunderstorm outflow boundary was modified after it propagated through a large wind farm. They found that the wind farm (>100 turbines) did slow down the outflow boundary; however, the effect on precipitation was minimal. In our study, it is unclear to what extent the effect of the turbines have on the propagation of the gravity waves. In response to a request by another reviewer, the SCADA data for additional rows has been included in the revised manuscript (Fig. 12) and the signal of the gravity waves is more notable in rows 3 and 4 in the SCADA data compared to the simulations. The simulations indicate that the farm weakens the gravity waves in the rotor layer; however, in the SCADA data, the waves maintain their strength throughout all four rows. This is likely because the observed gravity waves are more energetic compared to the simulated gravity waves. Please see lines 420-429 in the revised manuscript:

> "The simulations and SCADA data both show clear effects of the gravity waves in the power signal for row 1, but for rows 2-4 (especially rows 3 and 4), the gravity wave effects are more extreme in the SCADA compared to the simulation results. A potential reason for this difference is that the observed gravity waves could contain more energy than those predicted by the model. In the model, the gravity waves near the surface are dissipated as soon as they encounter the first row of the wind farm. In reality, the more energetic gravity waves are likely able to entrain momentum from above such that their effect is felt more strongly throughout all four rows. Considering the good agreement between the simulated and observed power signals for row 1, the model is able to capture the leading edge of the gravity wave passage but likely overestimates the dissipative effect of the wind farm on the gravity waves. Lastly, it is important to note that subtle

differences in the wind direction can cause large power fluctuations due to waking because of the configuration of the farm (as discussed in the following paragraphs)."

Figure 1: may help to have a larger map (perhaps on the scale of the state of Oklahoma) to provide some geographic perspective. I understand the need to capture the location of individual wind farms/turbines, but this is shown in Figure 4.

Figure 1 now includes a map of the continental United States with a red star highlighting the AWAKEN region within Oklahoma.

Table 3, and lines 365 et seq.: although it is demonstrated the gravity waves eliminated the LLJ, did the jet subsequently recover given there were several more hours until sunrise, and the "After" window only covers the period immediately after the waves have passed? This would also relate to the general vertical thermodynamic structure, and, of course, power production at the wind farm.

Thank you for the question. The jet did not recover before sunrise. Included below for the reviewer is a modified Fig. 7 extended up until well after sunrise (6:15am local time or 11:15 UTC). Only domain d01 was run for longer than the 'after' analysis period with the setup using the WDM6 microphysics scheme run the longest until 10:00am. Qualitatively, the results from domain d01 agree with the lidar observations with the jet not recovering much after the "after" period specified in the manuscript. The following sentence has been added to the manuscript at line 382-383:

> "However, in both observations and the model, the LLJ itself does not recover beyond the 'after' analysis period prior to sunrise (not shown)."

[Figure]

**References:**

Weckwerth, T. M. and Romatschke, U.: Where, When, and Why Did It Rain during PECAN?, Monthly Weather Review, 147, 3557 – 3573, https://doi.org/10.1175/MWR-D-18-0458.1, 2019.

Weckwerth, T. M., Hanesiak, J., Wilson, J. W., Trier, S. B., Degelia, S. K., Gallus, W. A., Roberts, R. D., and Wang, X.: Nocturnal Convection Initiation during PECAN 2015, Bulletin of the American Meteorological Society, 100, 2223 – 2239, https://doi.org/10.1175/BAMS-D-18-0299.1, 2019.

Tomaszewski, J. M. and Lundquist, J. K.: Observations and simulations of a wind farm modifying a thunderstorm outflow boundary, Wind Energ. Sci., 6, 1–13, https://doi.org/10.5194/wes-6-1-2021, 2021.

---

## Author Comment (AC3)

**Reviewer 1**

This is a well written paper which addressing a topic of interest which has been little explored. I would suggest that it can be published with some minor changes:

We thank the reviewer for the time they spent reviewing our work and for their supportive suggestions and comments. Our replies are inserted below into the reviewer's comments.

- Line 229: not sure what is meant by the 'bi-modal velocity deficit distribution'. Please elaborate.

We have clarified this sentence by referencing studies analyzing wind turbine wake characteristics and by specifying that the near-wake structure refers to the velocity deficit distribution prior to any turbulent mixing (lines 234-238 in the revised manuscript):

"When using 20~m grid spacing, and prior to the turbulent mixing in the far-wake region, the near-wake structure importantly still retains the characteristic bimodal distribution of the velocity deficit (also described as a double-Gaussian velocity deficit in Keane et al. (2016) and Schreiber et al. (2020)). The bimodal distribution is due to blade geometry and aerodynamics, as well as nacelle effects as seen in experiments, observations, and modeling (Wang et al., 2017; Vermeer et al., 2003; Carbajo Fuertes et al., 2018)."

- Figure 12b: it would useful to include the measured power output for rows 2-3 also. I understand that there may be commercial confidentiality issues, but presumably if the values are normalised as for row 1 there should not be a problem?

Figure 12 now includes the measured power output for rows 2-4. We have added the following discussion comparing and contrasting the simulated and measured power including all four rows on lines 420-429 in the revised manuscript:

"The simulations and SCADA data both show clear effects of the gravity waves in the power signal for row 1, but for rows 2-4 (especially rows 3 and 4), the gravity wave effects are more extreme in the SCADA compared to the simulation results. A potential reason for this difference is that the observed gravity waves could contain more energy than those predicted by the model. In the model, the gravity waves near the surface are dissipated as soon as they encounter the first row of the wind farm. In reality, the more energetic gravity waves are likely able to entrain momentum from above such that their effect is felt more strongly throughout all four rows. Considering the good agreement between the simulated and observed power signals for row 1, the model is able to capture the leading edge of the gravity wave passage but likely overestimates the dissipative effect of the wind farm on the gravity waves. Lastly, it is important to note that subtle differences in the wind direction can cause large power fluctuations due to waking because of the configuration of the farm (as discussed in the following paragraphs)."

[Figure]

- Figure 13: make it clear that the values are simulated (I presume)?

This has been clarified in the figure caption and at line 432 in the revised manuscript.

- Line 512: it is not obvious that there are patches with little or no turbulence in the TKE1.5 scheme which are much different to the other schemes. Maybe this could be highlighted on the plots?

An inflow region has now been highlighted in Fig. A1 and separately shown in Fig. A2. Along with the zoomed in plan slice of hub-height wind speed in Fig. A2, hub-height vertical velocity is also included to give the reader another visualization of turbulence that highlights the differences between the closures.

- Line 566: edit the superfluous text from the acknowledgements

The superfluous text has been removed.

There are a few typos:

Thank you for catching these mistakes. They have all been corrected.

- Figure 2 caption line 2: should be 'outlined', line 3 should be 'is outlined'

- Line 146: the Obukhov lengths should have units of metres
- Line 180: should be 'number of particle'
- Line 266: should be 'parameterization'
- Line 374: should be 'maximum' (I think?)
- Line 453: should be 'microphysics'
- Line 477: should be 'turbines are less...

**References:**

A Keane *et al* 2016 *J. Phys.: Conf. Ser.* **753** 032039 **DOI** 10.1088/1742-6596/753/3/032039

J Wang *et al* 2017 *J. Phys.: Conf. Ser.* **854** 012048 **DOI** 10.1088/1742-6596/854/1/012048

Schreiber, J., Balbaa, A., and Bottasso, C. L.: Brief communication: A double-Gaussian wake model, Wind Energ. Sci., 5, 237–244, https://doi.org/10.5194/wes-5-237-2020, 2020.

Carbajo Fuertes, F., Markfort, C. D., & Porté-Agel, F. (2018). Wind Turbine Wake Characterization with Nacelle-Mounted Wind Lidars for Analytical Wake Model Validation. *Remote Sensing*, *10*(5), 668. https://doi.org/10.3390/rs10050668

Vermeer, L.J., Sørensen, J.N. and Crespo, A. (2003) Wind Turbine Wake Aerodynamics. Progress in Aerospace Sciences, 39, 467-510. https://doi.org/10.1016/S0376-0421(03)00078-2